# Transient telomere uncapping triggers telomeric and subtelomeric rearrangements

Liébaut Dudragne[1,2], Clotilde Garrido[1,2], Oana Ilioaia[1,2], Juliana Silva Bernardes [ID][3✉] & Zhou Xu [ID][1,2✉]

## Abstract

**Telomeres cap the extremities of linear chromosomes and prevent their detection as DNA damage. Telomere uncapping poses a profound threat to genome integrity, yet the immediate consequences of transient uncapping remain unclear. In *Saccharomyces cerevisiae*, the Cdc13-Stn1-Ten1 complex limits resection, preventing DNA damage checkpoint activation. Here, using the temperature-sensitive *cdc13-1* allele, we demonstrate that transient telomere uncapping rapidly induces extensive genomic rearrangements despite a functional DNA damage checkpoint. Two distinct rearrangement signatures are observed in surviving cells: recombination of the subtelomeric region mostly involving the Y′ elements, and massively elongated telomeres up to 10 kb, a ~30-fold increase. Long-read sequencing evidences Y′ element losses/amplifications, terminal duplications, and telomeric-circle-driven amplifications of telomere repeats. Rearrangements unfold over multiple generations and require the homologous recombination factor Rad52, the Polδ subunit Pol32, and partially Rad51 and Rad59. Remarkably, survivors with elongated telomeres demonstrate a robust Rad52-dependent resistance to subsequent telomere uncapping. Our findings provide novel insights into the consequences of transient telomere uncapping for genome stability, a process that might contribute to subtelomere and telomere dynamics and evolution.**

**Keywords** Telomere; Subtelomere; Yeast; Genome Rearrangement; Long-Read Sequencing
**Subject Category** DNA Replication, Recombination & Repair

## Introduction

Protection of chromosome ends is critical for genome integrity. In most Eukaryotes, it is ensured by specialized repetitive DNA sequences, known as telomeres, that prevent chromosome ends from being recognized and processed as DNA damage (Jain and Cooper, 2010; de Lange, 2018). However, telomeres shorten at each cell division due to the end-replication problem. This process is compensated for by the recruitment of telomerase, an enzyme capable of elongating telomeres and maintaining them at a steady-state length distribution (Wellinger and Zakian, 2012). In *Saccharomyces cerevisiae*, telomeric sequences are composed of ~350 bp of degenerated $TG_{1-3}$ repeats with a short (5–15 nt) 3' single-stranded overhang (Larrivée et al, 2004; Soudet et al, 2014).

In yeast cells lacking telomerase, telomeres continuously shorten to eventually reach a critical length at which point cells activate the DNA damage checkpoint and stop dividing, in a process called replicative senescence (Lundblad and Szostak, 1989; Enomoto et al, 2002; IJpma and Greider, 2003). Genome instability, especially near chromosome ends, has been shown to increase during this phase (Coutelier et al, 2018; Hackett et al, 2001; Hackett and Greider, 2003). In prolonged telomerase-negative cultures, rare cells can escape senescence at an estimated frequency of ~$10^{-5}$ per generation by engaging in alternative, recombination-based pathways to maintain their chromosome ends (Lundblad and Blackburn, 1993; Chen et al, 2001; Le et al, 1999; Teng and Zakian, 1999; Teng et al, 2000; Kockler et al, 2021). These post-senescence survivors emerge through two distinct homology-dependent mechanisms: type I survivors amplify the subtelomeric Y' elements, whereas type II survivors elongate telomeric repeats, a mechanism also found in ALT cancer cells (Cesare and Reddel, 2010). Both types depend on Rad52, which is required for all types of recombination in yeast, and Pol32, a subunit of polymerase δ involved in break-induced replication (BIR) (Lundblad and Blackburn, 1993; Lydeard et al, 2007). Other genetic requirements for both types have been investigated in depth. For example, type I and type II survivors depend on the recombination factors Rad51 and Rad59, respectively (Teng et al, 2000; Chen et al, 2001). Although the type I and type II pathways have long been thought to be independent, a recent work proposed a unified pathway in which Rad51 is required for the formation of precursors, followed by Rad51-independent but Rad59-dependent maturation into stable survivors (Kockler et al, 2021). Thus, telomere shortening leads to genome instability and rearrangements in telomerase-independent survivors, which in turn bypasses the need for telomerase to maintain telomeres.

Telomere instability can also be induced when telomere protection fails. Telomeres are normally capped by the binding of specific proteins. Defects in these telomere-capping proteins or their binding have been shown to cause inappropriate repair events, such as telomere fusions or recombination involving telomeric or subtelomeric sequences (Fellerhoff et al, 2000; Garvik et al, 1995; Grandin et al, 2001; Grandin and Charbonneau, 2003; Marcand

[1]Sorbonne Université, CNRS, Laboratory of Computational, Quantitative and Synthetic Biology, CQSB, F-75005 Paris, France. [2]Sorbonne Université, CNRS, Inserm, Institut de Biologie Paris-Seine, IBPS, F-75005 Paris, France. [3]Sorbonne Université, CNRS, UMR 7144, Adaptation & Diversity in the Marine Environment, Station Biologique de Roscoff (SBR), 29680 Roscoff, France. ✉E-mail: juliana.silva_bernardes@sorbonne-universite.fr; zhou.xu@sorbonne-universite.fr

et al, 2008; Mieczkowski et al, 2003; Zubko and Lydall, 2006). These proteins include Rap1 and its cofactors Rif1, Rif2 and the Sir complex, the Ku complex, as well as the CST (Cdc13-Stn1-Ten1) complex (Wellinger and Zakian, 2012). Cdc13 is a key telomeric factor given its triple role in telomere maintenance: it assists telomere replication together with Stn1 and Ten1, promotes telomere elongation by recruiting telomerase and prevents excessive degradation of the 5' end by nucleases (Churikov et al, 2013). The *cdc13-1* allele is a hypomorph mutant leading to a temperature-sensitive phenotype widely used to study telomere capping (Garvik et al, 1995; Hartwell and Smith, 1985; Mersaoui and Wellinger, 2019; Paschini et al, 2012). The incubation of the *cdc13-1* mutant at restrictive temperatures (≥32 °C) leads to the resection of the 5' end and accumulation of single-stranded DNA (ssDNA) at telomeres, which activates the DNA damage checkpoint (Lydall and Weinert, 1995; Garvik et al, 1995). Permanent incubation of *cdc13-1* cells at restrictive temperature leads to cell death, despite a limited number of cell divisions allowed by adapting to the checkpoint in the first 24 h (Lee et al, 1998; Sandell and Zakian, 1993; Toczyski et al, 1997). Cells resistant to permanent telomere uncapping can only be selected if the resection or checkpoint pathways are also altered (Grandin et al, 2001; Grandin and Charbonneau, 2003, 2013; Zubko and Lydall, 2006), suggesting that cells with deprotected telomeres are eliminated in conditions with functional DNA processing and checkpoint, thus safeguarding genome integrity. For example, checkpoint-deficient *mec3Δ cdc13-1* cells incubated at an intermediate temperature of 29 °C tolerated telomere deprotection and, after several passages, generated Cdc13-independent survivors with very long telomeres akin to those of telomerase-negative type II survivors (Grandin et al, 2001). Similar survivors were also generated in *cdc13-1 exo1Δ* mutants grown at 36 °C, and at a higher frequency in the checkpoint-deficient *cdc13-1 exo1Δ rad9Δ* mutant (Zubko and Lydall, 2006).

Overall, Cdc13 dysfunction alone, without additional mutations in checkpoint- or resection-related genes, has not been shown to lead to the emergence of cells with rearranged telomere structures. In addition, previous studies of the long-term consequences of permanent telomere deprotection might have missed the direct molecular response to telomere deprotection. Here, instead, we investigated the early effects of telomere deprotection on genome stability in the presence of a functional checkpoint. To do so, we use an experimental protocol where telomeres are transiently deprotected, and cells are allowed to recover, thus enabling us to characterize the full spectrum of molecular consequences, including events that would have been counterselected if telomeres were kept deprotected permanently. We find that genomic instability arises earlier than anticipated, even in the presence of a functional checkpoint, and is limited to the telomere and subtelomere regions. Using long-read sequencing and telomere-to-telomere genome assembly, we characterized Rad52- and Pol32-dependent rearrangements of the subtelomeric Y' elements and telomere elongation up to 8–10 kb. Strikingly, we discovered that very long telomeres together with Rad52 protect cells from a second telomere uncapping event, suggesting that this telomeric nucleoprotein structure acts as a Cdc13-independent protective cap. Our results provide new insights into telomere-driven genomic instability and underscore the impact of transient telomere dysfunction on chromosome integrity.

# Results

## Transient telomere uncapping impairs cell survival

To evaluate the direct molecular and cellular response to telomeric uncapping, we employed the temperature-sensitive *cdc13-1* allele to induce a transient uncapping. We exposed the *cdc13-1* strain to the restrictive temperature (32 °C) for various durations. The cells were first grown overnight in rich liquid medium at the permissive temperature (23 °C), then plated on solid YPD media and incubated at 32 °C for a defined time "*t*" (0–72 h) to induce telomere uncapping (Fig. 1A). The cells were then returned to the permissive temperature to assess recovery and survival. The cells tolerated uncapping for up to 6 h without a noticeable decrease in viability, but the survival frequency decreased significantly for $t \geq 12$ h (Fig. 1B,C). Some rare colonies formed even after uninterrupted incubation at 32 °C for 72 h, establishing a lower limit for cell survival to *cdc13-1*-induced telomeric uncapping at 32 °C. The survival frequency for $t = 24$ h was 5.5-fold greater than for $t = 72$ h (Fig. 1C).

We conclude that transient Cdc13 dysfunction at telomeres impairs cell survival in a time-dependent manner. To investigate the molecular changes induced by transient uncapping, we selected the 24-h-long uncapping condition for further analysis.

## Cells exposed to telomere uncapping for 24 h show extensive genomic rearrangements

To study the impact of transient *cdc13-1*-mediated uncapping on genomic stability, we selected and grew colonies formed from cells that survived 24 h at 32 °C. We investigated large-scale genomic rearrangements in these survivors using pulsed field gel electrophoresis (PFGE), which enables the resolution of whole chromosomes on agarose gels. Among the seven analyzed colonies, six presented significant shifts in chromosome length, corresponding to changes of at least tens of thousands of base pairs (Fig. 2A).

Regions adjacent to the telomeric repeats typically contain Telomere-Associated Sequences (TASs), namely X and Y' elements. Y' elements, present in zero, one, or multiple tandem copies separated by short interstitial telomeric sequences, exist in two main size classes: long (6.7 kb) and short (5.2 kb). The X element, which is found at all extremities in laboratory strains, varies in sequence and size. To test whether telomeres or TASs are involved in the observed instability, we employed terminal restriction fragment (TRF) Southern blotting, which probes telomere sequences in *XhoI*-restricted fragments to inform on characteristic features of the telomere structure. In the control *cdc13-1* strain grown at 23 °C, telomeres associated with XY' extremities ("Y' telomeres" in Fig. 2B) migrated to form a smear near 1.2 kb, corresponding to ~300 bp of telomeric repeats and a ~900 bp segment defined by the *XhoI* site in the Y' element. In contrast, telomeres at X-only extremities ("X fragments" in Fig. 2B) migrated as longer fragments owing to more distal *XhoI* restriction sites. Additionally, interstitial telomere sequences between Y' elements allowed probing of tandem Y' repeats ("Y' fragments" in Fig. 2B). Out of 31 clones that survived transient telomere uncapping, 6 exhibited very long and heterogeneous telomeres (c8, c12, c13, c20, c21, and c27, in Figs. 2B and EV1A), a pattern that resembles that observed in telomerase-independent, recombination-dependent,

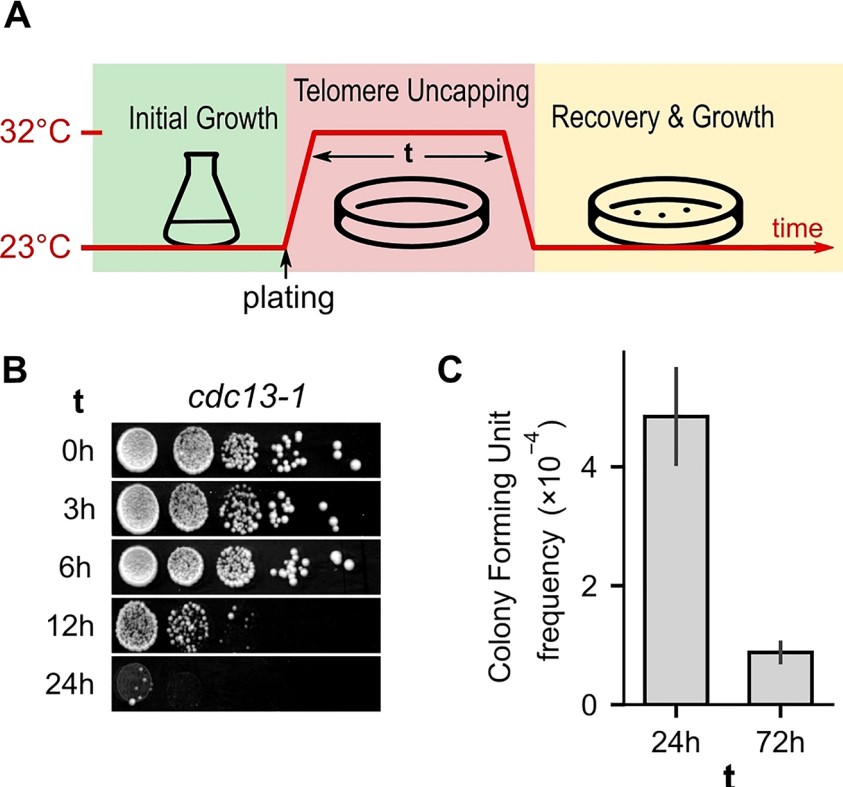

**Figure 1. Effect of transient telomere uncapping on survival.**

(A) Scheme of the experimental protocol: *cdc13-1* cells were first grown at permissive temperature (23 °C) before plating and incubation at restrictive temperature (32 °C) for a duration *t*. The plate was then returned to 23 °C for recovery and growth. Individual colonies were picked for further analysis. (B) Spot assay showing the growth of *cdc13-1* cells on YPD rich media after *t* hours of telomere uncapping. (C) Colony formation frequency measured using the protocol shown in (A). A control plate was maintained at 23 °C for normalization. The error bars correspond to the standard error of the mean; $n = 7$ (24 h) and $n = 4$ (72 h) independent experiments. Source data are available online for this figure.

type II survivors (Lundblad and Blackburn, 1993). The majority of other survivors showed visible loss of X fragments, and/or alterations in Y′-fragments intensity when marked with a Y′-specific probe (Figs. 2B and EV1A,B). These are indicative of putative Y′ acquisition at X-only subtelomeres or Y′ copy number alterations, respectively. Y′ amplification and acquisition are also observed in type I post-senescent survivors, although to a much greater extent, where no X band is retained and the Y′ signal is strongly amplified. Notably, we observed no telomere shortening in any clone, which is consistent with telomerase being active (Fig. 2B). We referred to these uncapping-induced rearrangement patterns as type-II-like (T-II-L) survivors for the survivors with lengthened telomeres and Y′-associated survivors (YAS) for those only exhibiting altered Y′ organization.

## Genomic instability unravels over multiple cell divisions independently of checkpoint adaptation

The PFGE profiles of several clones presented blurred bands or bands of altered relative intensities, suggesting intraclonal heterogeneity (Fig. 2A). To test this possibility, three primary clones were subcloned, and four subclones of each were reanalyzed by PFGE. All three primary clones exhibited subclonal heterogeneity in their

PFGE patterns, revealing that genomic instability persisted during several cell divisions (Figs. 2C and EV1C). This was particularly obvious in clone c3, where no two subclones shared the same pattern, indicative of at least two rounds of divisions with ongoing rearrangements. Whether these cell divisions occurred while telomeres were still uncapped or after returning to 23 °C was unclear.

Indeed, after prolonged activation (~8 h) of the DNA damage checkpoint, cells can undergo checkpoint adaptation (Sandell and Zakian, 1993; Toczyski et al, 1997; Lee et al, 1998), resuming cell cycle despite unrepaired DNA damage and allowing for a few cell divisions while telomeres are uncapped. This process is associated with increased genomic instability (Galgoczy and Toczyski, 2001; Coutelier et al, 2018). To assess the role of checkpoint adaptation both in genomic instability and in intraclonal heterogeneity, we introduced the adaptation-deficient *cdc5-ad* allele of *CDC5*, encoding the Polo kinase essential for late cell cycle events and for adaptation (Toczyski et al, 1997). *cdc5-ad cdc13-1* cells that survived 24 h of telomere uncapping exhibited levels of genome rearrangements and intraclonal heterogeneity comparable to the *CDC5 cdc13-1* strain (Fig. EV1D,E). Since *cdc5-ad cdc13-1* cells do not divide while at 32 °C, this finding indicates that genomic instability could persist over several divisions after telomere

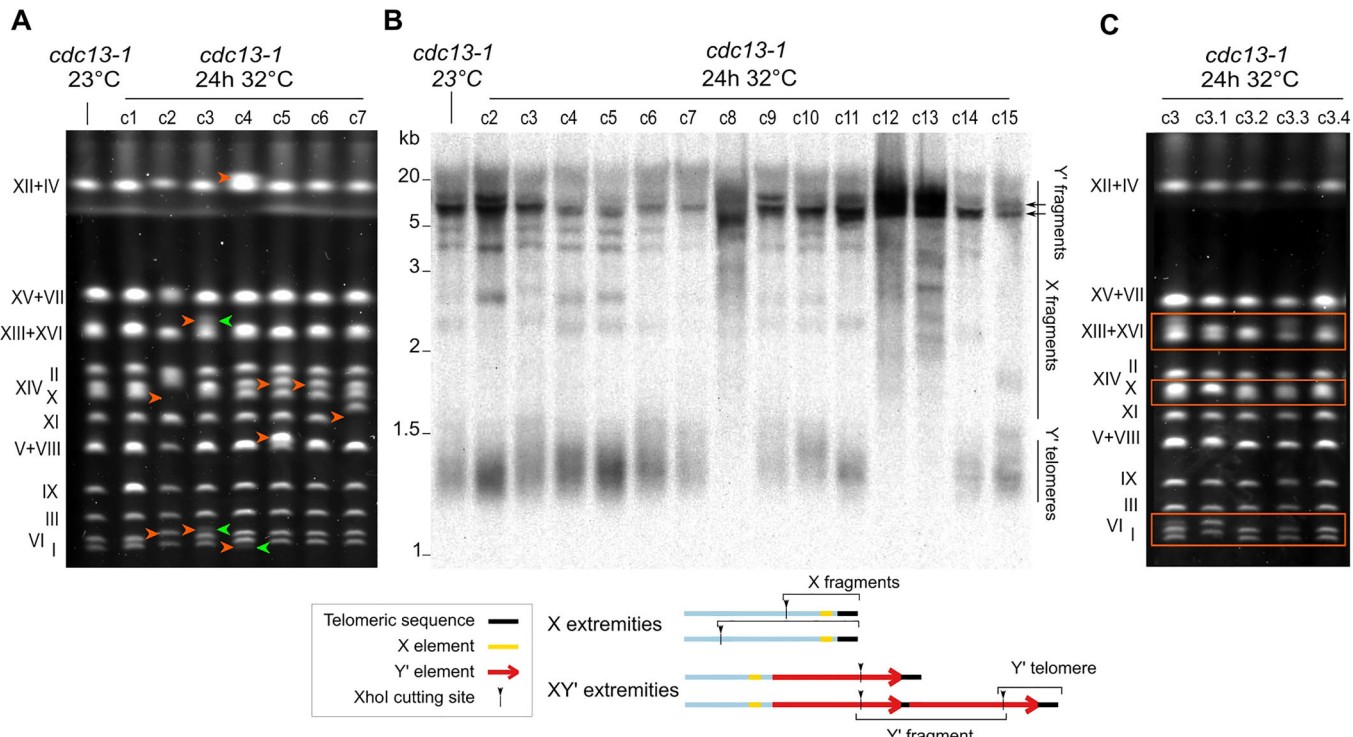

**Figure 2. Transient telomere uncapping leads to chromosomal instability and changes in telomere and subtelomere structure.**

(A) PFGE of 7 *cdc13-1* survivor clones, as well as a *cdc13-1* control strain grown constantly at 23 °C. Compared to the control strain, six out of the seven survivor clones exhibited apparent chromosome size shifts, marked by orange arrows. Blurred bands or bands of altered intensity indicative of probable population heterogeneity are indicated by green arrows. (B) TRF Southern blot analysis of 14 *cdc13-1* survivor clones and a control strain. The genomic DNAs were cut with the *XhoI* enzyme, and a telomeric probe was used. A schematic representation of subtelomere structures and *XhoI* restriction fragments is displayed below the blot. (C) PFGE of 4 subclones of clone 3 from (A, B), derived from single cells grown at 23 °C. The orange boxes indicate bands with distinct migration patterns in the 4 subclones. See Fig. EV1C for two other clones and their respective subclones, in addition to this one. Source data are available online for this figure.

protection is re-established. To circumvent this intraclonal heterogeneity, all subsequent analyses were conducted using subclones of survivor colonies.

## Genomic rearrangements are confined to telomeric regions

Given the important size shifts observed on the PFGE gels, we wondered whether the observed genomic instability was confined to the telomeric regions. To comprehensively assess the localization of the rearrangements, we performed Oxford Nanopore long-read sequencing on the genomic DNA of 9 survivors as well as the *cdc13-1* control strain, which did not experience telomere uncapping. Three T-II-L clones were selected from the second, better-resolved Southern blot (Fig. EV1A), and six YAS clones were randomly selected to be representative of both altered and unaltered Southern profiles. The longest and highest quality nanopore reads were assembled, and the assemblies were manually inspected to correct any assembly mistakes in telomeric regions. This procedure yielded highly contiguous, high-quality telomere-to-telomere genome assemblies (Table EV2).

Several genetic screens identified gene mutations that partially suppress the growth defect of *cdc13-1* at restrictive, semi-permissive, or oscillating temperatures (Addinall et al,

2008, 2011; Holstein et al, 2017). We thus tested whether the survivor clones we selected could have mutated any of these genes. We performed a variant calling on these 9 genome assemblies and compared the detected mutations with the gene lists recovered from the three genetic screens (Addinall et al, 2008, 2011; Holstein et al, 2017). Mutations in only 25 genes were found in at least two genomes, and only one gene, *MTC7*, was previously reported (Addinall et al, 2008). However, a mutation of *MTC7* was reported to be associated with decreased survival in combination with telomere uncapping. Thus, we did not find any evidence of selection of common mutations promoting growth in the context of telomere deprotection in either YAS or T-II-L survivors.

To investigate the rearrangements in telomeric regions, we first focused on the dynamics of Y' elements. To track them accurately, Y' elements identified in the control strain genome assembly were labeled and clustered based on sequence similarity (Fig. 3A). We identified 34 Y' elements found at 20 out of 32 extremities, with 0 to 6 tandem Y' elements per extremity (Dataset EV1). Among the 34 Y' elements, we detected 20 unique sequence variants that could be grouped into 11 clusters. Two elements were classified as the same variant if their sequences were identical within the expected margin of assembly errors. Clustering analysis recovered the two main size families of Y' elements—short and long—along with finer sub-clusters, revealing the presence of outliers alongside large groups

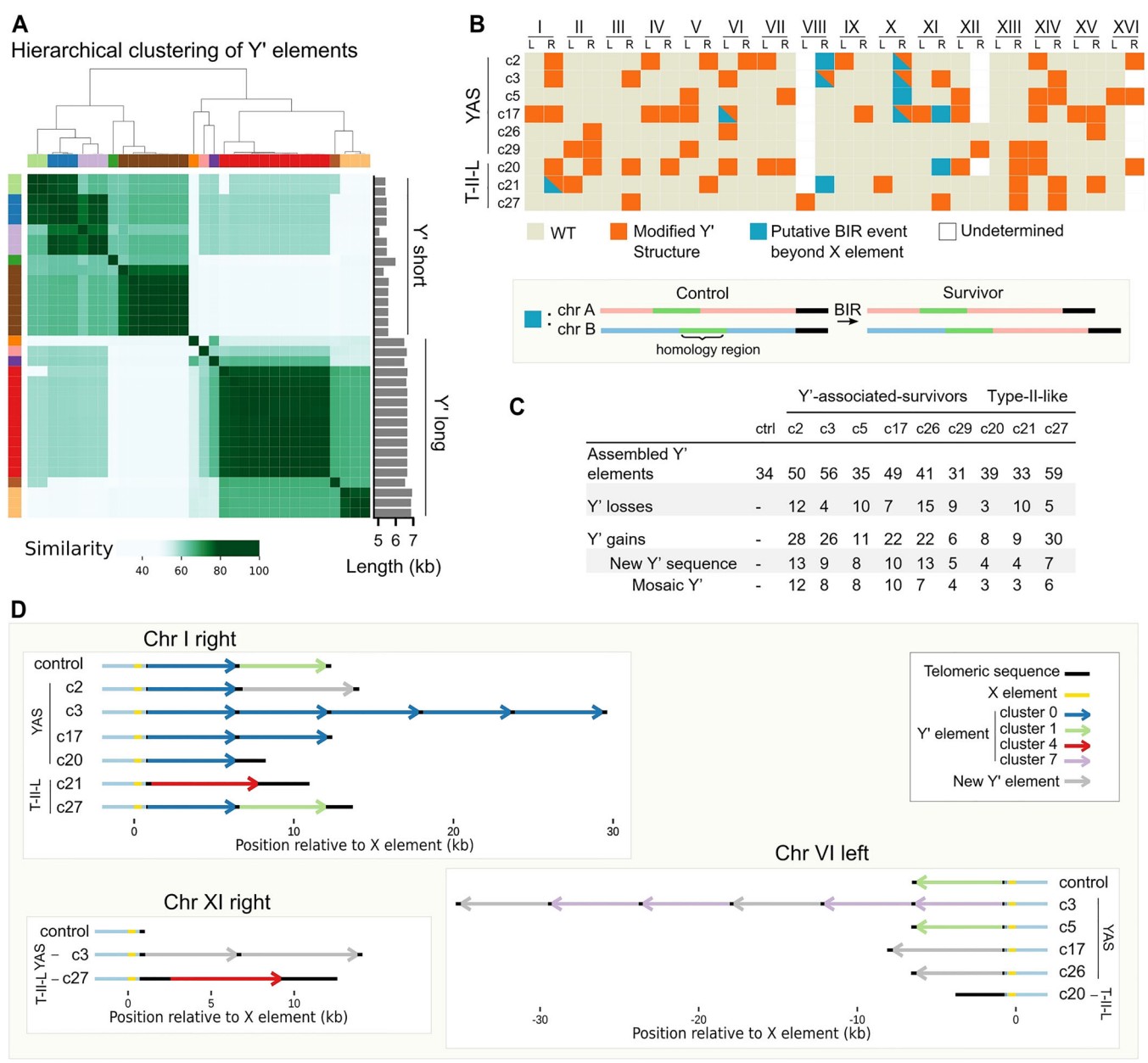

Figure 3. Subtelomeric Y′ elements are rearranged following telomere uncapping.

(A) Similarity matrix and clustering of Y′ elements of the control strain. A similarity measure was computed for each pair of Y′ sequences and used for subsequent hierarchical clustering (dendrogram shown above the matrix). The colors correspond to the different clusters (n = 11 clusters). The bar plot on the right side indicates the corresponding lengths of the Y′ sequences. See Dataset EV1 for the sequences of all 34 Y′ elements. (B) Map of alterations at all chromosome extremities. The labeling of Y′ elements was used to detect changes of Y′ structure at a given telomere (orange square or triangle), and duplications in core genome sequences indicate probable BIR events (blue square or triangle). YAS Y′-associated survivors, T-II-L type-II-like survivors. Extremities with more than 5 Y′ elements in the reference could not always be resolved with high confidence and are thus marked as "undetermined". A similar map with rearrangement subcategories is shown in Fig. EV3D. (C) Summary table of the total number of indicated Y′ changes in survivor clones. (D) Visual representation of an illustrative subset of modified Y′ structures. Three extremities are displayed as examples of typical modifications of subtelomere structure found in the indicated clones, as compared with the control: Y′ losses, gains, tandem duplications, and new Y′ sequence emergence. The long telomere sequences of type-II-like survivors (c20, c21, and c27) are apparent. See Dataset EV2 for the representation of all extremities across all sequenced clones.

(Fig. 3A). Interestingly, although Y′ elements are typically classified as either 5.2 kb or 6.7 kb in size, we observed that Y′ short elements ranged from 5062 bp to 5592 bp (mean: 5470 bp), Y′ long elements ranged from 6484 bp to 6933 bp (mean: 6670 bp), and one element exhibited an intermediate size of 5981 bp.

Using this labeling and clustering approach, we quantified the number of Y′ elements associated with rearrangement events in the clones surviving transient uncapping (Fig. 3B,C). On average, 8.3 ± 3.7 (mean ± SD) Y′ elements were lost, and 18 ± 8.9 were gained in each survivor clone, affecting 7.4 ± 3.1 extremities. These

numbers far exceeded what could be confidently detected by TRF Southern blot analysis, especially considering c3 and c5, whose profiles seemed unaffected both in telomere-probed and Y'-probed membranes (Figs. 2B and EV1A,B). These events included loss or gain of a known Y' element, tandem amplification of existing elements, and most interestingly, emergence of new Y' elements whose sequences did not correspond to any element present in the control (Figs. 3C,D; Dataset EV2). The frequency of these Y' structure alterations was significantly higher at Y' telomeres than at X telomeres (28 vs. 15%; p value = 0.007, Mann–Whitney test). We found that at least 84% of the new Y' elements could be explained as mosaics of 2 or more other Y' elements, suggesting that most of them originated from recombination events between existing Y' elements rather than mutation accumulation. Interestingly, type-II-like survivors exhibited levels of Y' rearrangements comparable to Y'-associated survivors (YAS: 7.3 ± 3.3 extremities affected, T-II-L: 7.7 ± 2.5 extremities affected; mean ± SD).

We also examined whether genome rearrangements extended beyond the X and Y' elements into more centromere-proximal regions. We found ten cases of terminal duplications from one chromosome end to another, spanning between 850 bp and 89 kb of sequences centromere-proximal to the X element across five clones (Fig. 3B). In two cases, two events were found at the same extremity, with one duplication contained within the other. By mapping these regions against the control genome, we found that these events consistently involved junctions in regions of homology between the donor and recipient chromosome arms, the size of which ranged from several hundreds to several thousands of base pairs. Based on the presence of homology and on the pattern of terminal duplication, these events likely resulted from break-induced replication (BIR).

Finally, we used the variant callers Sniffles2 (Smolka et al, 2024) and PAV (Ebert et al, 2021) to identify other structural rearrangements (>50 bp) that might have occurred in the rest of the genome. No event was detected aside from those described above, indicating that genome rearrangements were confined to telomere-proximal regions.

## Type II-like survivors exhibit tandem repeat amplification of telomeres

To study the impact of uncapping on telomere sequences, telomere length distributions were measured from the sequencing reads of all sequenced survivors (Garrido et al, 2026). In Y'-associated survivors, no significant change in telomere length distribution was observed in any clone (Fig. 4A), which is consistent with the results of TRF Southern blot analysis (Figs. 2B and EV1A). In contrast, we confirmed that type-II-like strains presented very heterogeneous and long telomeres of up to 10 kb (Figs. 4B,C and EV2A–C). Telomere length distributions in type-II-like survivors presented multiple peaks, explained by the heterogeneity in chromosome extremity-specific average lengths, including a major peak aggregating 4–7 extremities with an average telomere length <500 bp (Figs. 4C and EV2B,C), indicating that not all telomeres were elongated equally.

Telomere lengthening in telomerase-negative post-senescence survivors is attributed to recombination events and rolling circle amplification of telomeric circles (t-circles) (Natarajan and McEachern, 2002; Lin et al, 2005; Larrivée and Wellinger, 2006;

Aguilera et al, 2022). Telomerase in budding yeast adds imperfect repeats and creates a degenerated telomere motif (Forstemann, 2000). As a consequence, long enough telomere sequences generated by telomerase are unique and can be identified unambiguously. In contrast, recombination between telomeres or rolling circle amplification is expected to yield 2 or more copies of identical sequences. We thus looked for tandem amplification of telomeric sequences. In the control strain, we found a few tandem sequences of two copies for repeats of < 45 bp (Fig. 4D). These tandem repeats could either be signatures of previous events of recombination at telomeres or instances in which telomerase stochastically added two identical sequences in tandem. To specifically capture signatures of telomere uncapping, we excluded all repeats shorter than 50 bp from our analysis of type-II-like survivors. We found that the three sequenced type-II-like strains contained many perfect tandem repeats at telomeres. While we detected relatively short sequences (<100 bp) that were repeated three times or fewer in tandem, some repeats were longer than 600 bp and others were found in arrays of up to nine copies (Fig. 4D). We found multiple raw nanopore reads supporting these tandem repeats, thus excluding assembly errors as a potential explanation for these observations. Tandem repeats found in at least one array of three or more copies were then selected for more exhaustive detection. This second step allowed the detection of single and slightly imperfect occurrences of these same repeats. Interestingly, several of the same repeated sequences were found at multiple telomeres, often beginning at different positions within the repeat, strongly suggesting that these tandem repeats might have resulted from t-circle-mediated telomere amplification (Figs. 4E and EV2D,E). The exact sequence of one circular molecule (Circle_139 from clone c21) was found in the genome of the control strain grown at 23 °C, on the telomere of chromosome V left. This finding suggested that circular molecules might have been excised from endogenous telomeres. The origin of other circles could not be identified in the genome of the control strain, likely because the sequence of telomere termini is highly divergent due to telomerase activity. Additionally, the 139-bp-long circle sequence was contained within the 332-bp-long one in the same survivor clone, suggesting that the latter might have derived from a secondary excision event after the former was copied into another telomere. We could not find sequence identity between any other pair of circle sequences within a clone, suggesting independent origins.

Depending on the clone, between 13% and 32% of the total telomeric DNA was part of perfect tandem repeats, values that could increase up to between 23% and 44% when taking single or imperfect repeats into account. While it could not be excluded that some of these repeats were further propagated through telomere-to-telomere recombination, these numbers highlight the importance of the inferred t-circle-mediated amplification in response to telomere uncapping.

## Telomere-uncapping-induced rearrangements are RAD52- and POL32-dependent

The Y' rearrangement events and telomere recombination signatures we observed suggest a key role of homologous recombination (HR) pathways. To investigate the involvement of HR factors, we deleted their corresponding genes in the cdc13-1 strain (Fig. 5A–D),

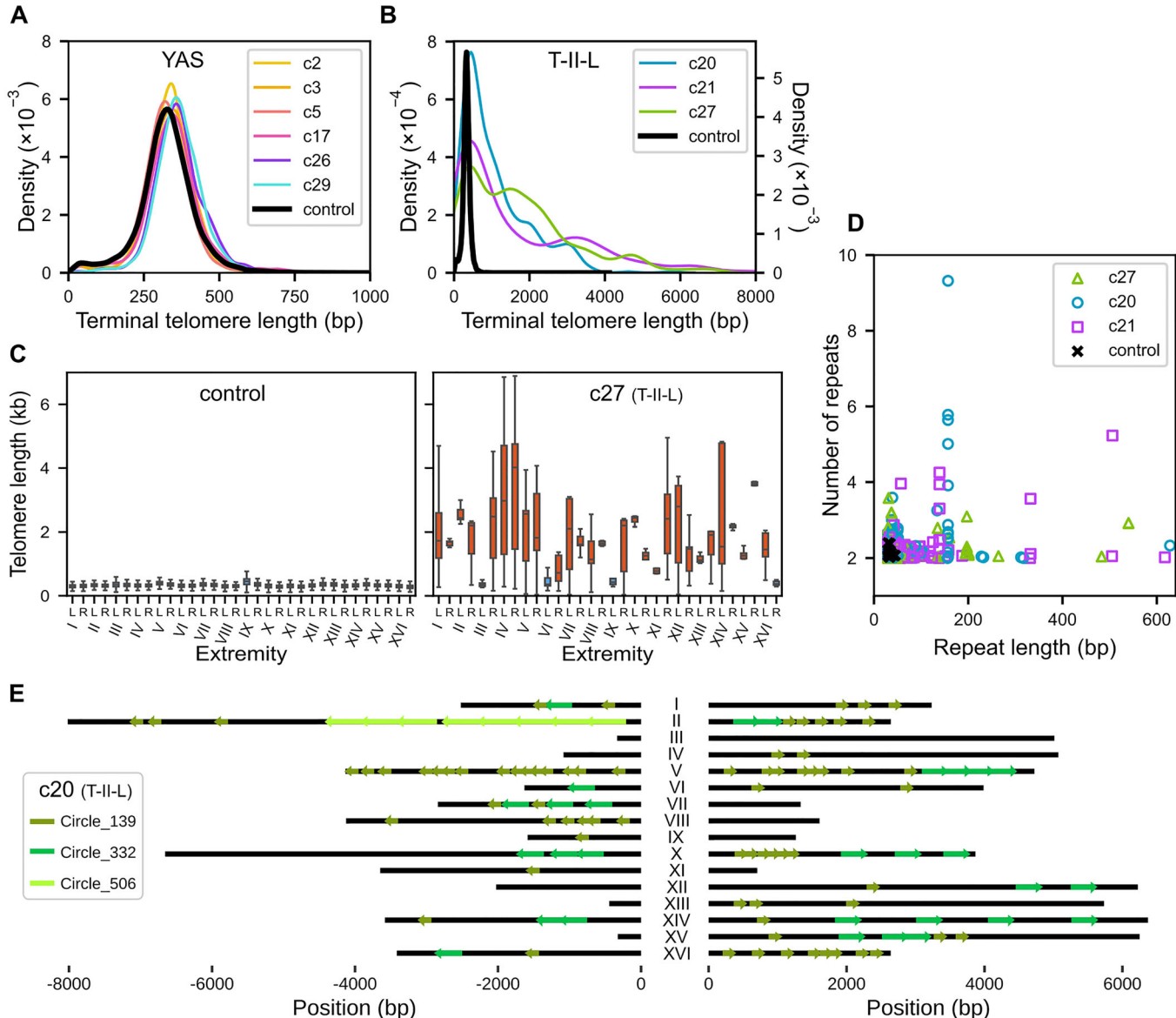

**Figure 4. Telomeres are elongated heterogeneously in T-II-L survivors and contain large perfect tandem repeats.**

(A) Telomere length distributions of YAS strains compared to the control. (B) Telomere length distribution of T-II-L survivors. The left y-axis is used for survivors, whereas the right y-axis is used for the control strain. The x-axis is truncated at 8000 bp. (C) Boxplots representing the telomere length distribution at each chromosome extremity. The box is bounded by the first quartile (Q1) and the third quartile (Q3), with the median indicated by a line, while the whiskers extend to $Q1 - 1.5 \times IQR$ and $Q3 + 1.5 \times IQR$, with IQR being the interquartile range Q3-Q1. The number of reads used to draw the boxplots is comprised between $n = 180$ and $n = 352$ for the control and $n = 5$ and $n = 31$ for survivor c27. The control strain (left) displays extremity-specific mean telomere lengths of ~300–400 bp with limited variations between extremities (see also Fig. EV2A), whereas survivor c27 shows heterogeneously elongated telomeres with great variations between extremities (see Fig. EV2B,C for other T-II-L survivors). The blue boxplots correspond to distributions with a mean telomere length <500 bp. (D) Arrays of perfect tandem telomere repeats detected in the genome assemblies of T-II-L survivors and the control strain are represented in a 2D plot showing the number of repeats and the length of each repeat. (E) Visualization of putative circle-derived sequences detected in the left and right telomeres of the 16 chromosomes of the T-II-L clone c21. The circle number corresponds to the length of the circle sequence. Black horizontal lines represent the assembled telomeric sequences. (see Fig. EV2D, E for other T-II-L survivors).

starting with *RAD52* and *POL32*. We exposed *cdc13-1 rad52Δ* and *cdc13-1 pol32Δ* cells to 24 h of telomere uncapping followed by recovery at 23 °C. A mild ~2-fold decrease in survival in *cdc13-1 rad52Δ* and a ~10-fold decrease in survival in *cdc13-1 pol32Δ* compared to wild-type were observed, in contrast to the deletion of *EXO1*, which led to a ~60-fold increase in survival (Fig. EV3A). Following a milder 12-h transient uncapping, the deletion of

*RAD52* had no significant effect on survival, in contrast to *POL32*, suggesting a potential genetic interaction between *POL32* and *cdc13-1* (Fig. EV3B). TRF Southern blot analysis showed that none of the *rad52Δ* and *pol32Δ* survivor clones presented either Y'-associated rearrangements or massive telomere elongation expected in type-II-like survivors (Fig. 5A,B). We then performed nanopore sequencing and genome assembly on survivor clones of both

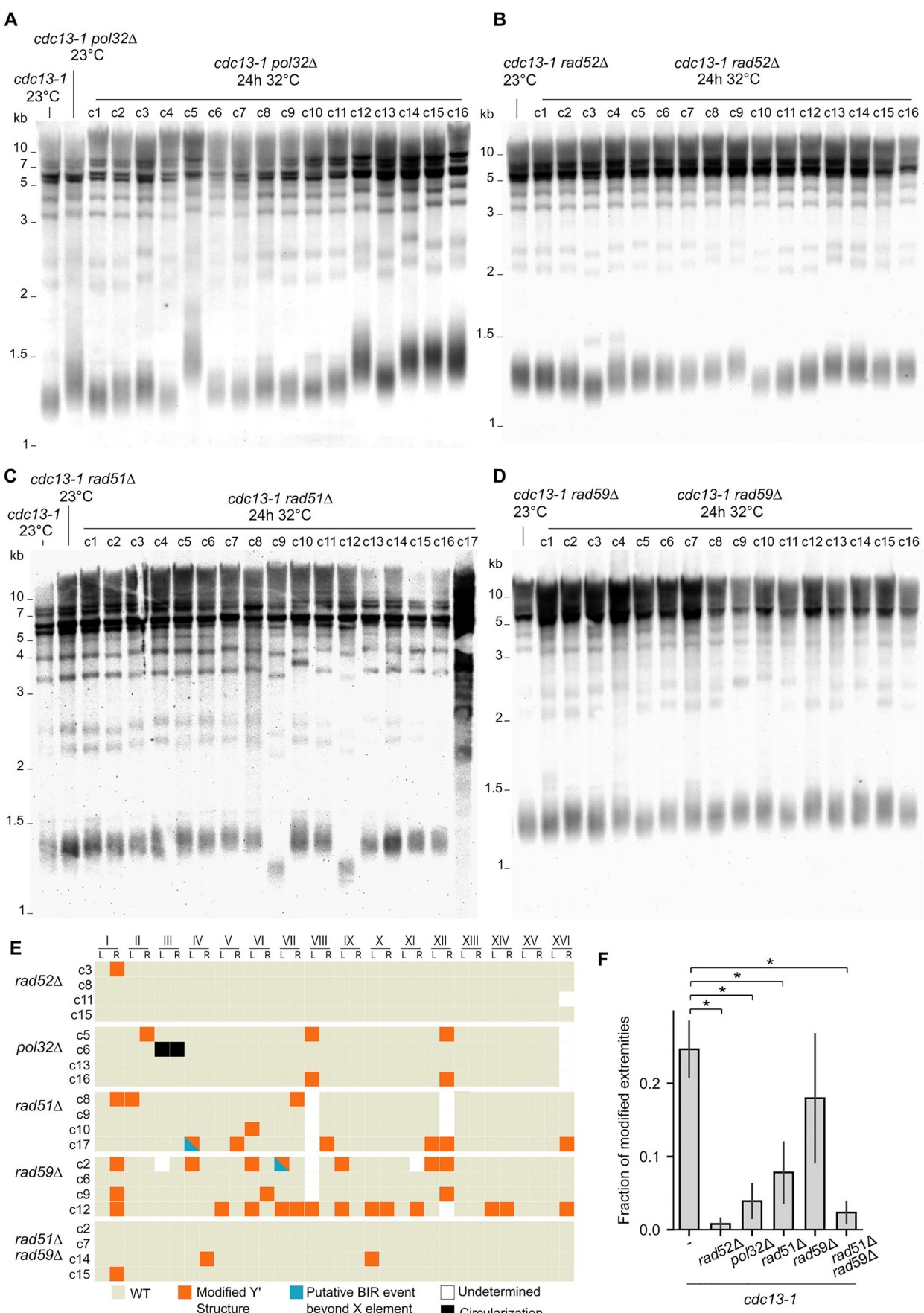

**Figure 5.** *RAD52* and *POL32* are essential for telomere-uncapping-induced rearrangements, while *RAD51* and *RAD59* are partially required.

(A–D) TRF Southern blots of *cdc13-1 pol32Δ* (A), *cd13-1 rad52Δ* (B), *cdc13-1 rad51Δ* (C), and *cdc13-1 rad59Δ* (D) survivor colonies. Cultures from the same strains that were maintained at 23 °C were used as controls. (E) Map of chromosome extremity alterations as in Fig. 3B, for sequenced survivor clones of the indicated genotype. A chromosome circularization event in one *pol32Δ* survivor is indicated by black squares. A similar map with rearrangement subcategories is shown in Fig. EV3D. See Dataset EV2 for the representation of all extremities across all sequenced clones. (F) Fractions of extremities with altered sequence or structure in the indicated strains ($n = 9$ for the wild-type *cdc13-1* strain and $n = 4$ for all mutants, error bars represent the standard error of the mean). A statistically significant decrease was observed for the *rad52Δ*, *pol32Δ*, *rad51Δ* and *rad51Δ rad59Δ* mutants (Student's *t*-test, $_*$: from top to bottom, *p* value = 0.0031, 0.023, 0.0056, and 0.0019). Source data are available online for this figure.

genotypes to better characterize potential rearrangements that might have escaped detection by Southern blot. The genomes of the *cdc13-1 rad52Δ* and *cdc13-1 pol32Δ* survivor clones almost perfectly matched the control strains in telomeric regions (Figs. 5E,F and EV3D). The only alteration affecting Y' elements in the *rad52Δ* strain was the loss of a single Y' element. In one *pol32Δ* survivor clone, we found that chromosome III circularized using *HML* and *HMR* for homology. Circularization of chromosome III has previously been described in the context of mating-type interconversion events and recombination between mating-type loci (Strathern et al, 1979; Klar et al, 1983). Interestingly, while the *cdc13-1 rad52Δ* control strain and survivors showed modest variations in telomere length, telomeres in *cdc13-1 pol32Δ* were already longer at 23 °C, consistent with a previous report (Gatbonton et al, 2006), and telomere-uncapping-induced survivors displayed highly variable telomere lengths (Figs. 5A,B and EV4A,B). Overall, these results indicate that telomeric and subtelomeric rearrangements strongly depend on Rad52 and Pol32. However, the fact that the survival to transient telomere uncapping is partially independent of Rad52 suggests that the rearrangements are a consequence of transient telomere uncapping and, although they might contribute to survival, they are not absolutely required.

## Rad51 and Rad59 are partially involved in Y' recombination

To further dissect the molecular pathways underlying Y'-associated and type-II-like rearrangements, we tested the roles of Rad51 and Rad59. We exposed the *cdc13-1 rad51Δ* and *cdc13-1 rad59Δ* mutants to transient telomere uncapping and observed a ~2-fold decrease in survival in both mutants, further suggesting a protective role of homologous recombination at uncapped telomeres (Fig. EV3A).

Next, we examined the structural changes in survivors using the TRF Southern blot. Only 1 out of 27 *rad51Δ* survivors (c17) and 0 out of 16 in *rad59Δ* survivors displayed a type-II-like pattern (Fig. 5C,D). While the sample size limited the statistical power, these results suggest the involvement of Rad51 and Rad59 in type-II-like formation (*p* value = 0.11 and 0.08, respectively, Fisher's exact test). Furthermore, the effects of both proteins became clearer when type-II-like signature and X band alterations were considered simultaneously: in *cdc13-1*, 18 out of 30 clones displayed alterations in TRF Southern blot, whereas only five out of 26 *cdc13-1 rad51Δ* clones (*p* value = 0.002, Fischer's exact test) and 4 out of 16 in *rad59Δ cdc13-1* clones (*p* value = 0.03, Fischer's exact test) were affected. Additionally, nanopore sequencing and analysis of genome assemblies of a subset of clones confirmed a strong

decrease in Y' recombination frequency in *rad51Δ*, whereas the effect in *rad59Δ* appeared milder (Fig. 5E,F). Telomeric sequences analysis of the T-II-L *rad51Δ* survivor (c17) revealed similar tandem sequence patterns as found in the wild-type T-II-L, with one putative circle (Circle_58) being successfully traced back to the left telomere of chromosome XIII (Fig. EV4E). Moreover, the 58-bp circle sequence was present within all other circles from this strain, except for the 124-bp-long one.

We then constructed the *rad51Δ rad59Δ* double mutant and found a very low frequency of modified subtelomeres, which was lower than that of each single mutant and at a level similar to that of *rad52Δ*, based on TRF Southern blot analysis and nanopore sequencing (Figs. 5E,F and EV3C). Most remaining Y' alterations were Y' losses, which did not necessarily depend on recombination. In contrast, only 2 out of 10 Y' alterations in *rad51Δ* were losses (Fig. EV3D).

Intriguingly, two *rad51Δ* survivors exhibited a marked reduction in telomere length (Figs. 5C and EV4C). Telomere length distribution analysis of clone c9 revealed a decrease from the expected 350 bp to ~200 bp across all chromosome ends (Fig. EV4C), suggesting a potential change in global telomere length regulation in this survivor. In contrast, the telomere length distributions of the *rad59Δ* and *rad59Δ rad51Δ* survivors were not altered compared to the control (Figs. 5D and EV3B and EV4D).

Taken together, these findings suggest that Rad51 and Rad59 both contribute to the telomeric and subtelomeric rearrangements found in Y'-associated and type-II-like survivors, and function in at least partially independent pathways.

## Type-II-like survivors are resistant to a second telomere uncapping

To test whether recombination at telomeres and subtelomeres allows yeast cells to resist telomere uncapping, we challenged four Y'-associated survivors and three type-II-like survivors with a second 24-h incubation at 32 °C. While both survivor types grew normally at 23 °C, only the Y'-associated survivors displayed a loss of viability comparable to that of the original *cdc13-1* control after the second stress (Fig. 6A). In contrast, all three type-II-like survivors maintained robust growth after 24 h at 32 °C, showing little to no survival defect. We wondered whether the DNA damage checkpoint pathway in type-II-like survivors might be dysfunctional, thus supporting cell growth despite telomere uncapping. We measured the fraction of unbudded G1 cells in type-II-like survivor cells after 3 h at 32 °C and found it decreased to a similar extent as in the initial *cdc13-1* strain, as quantified by microscopy (Fig. EV5A), suggesting that the DNA damage checkpoint in response to telomere uncapping was still functional in type-II-like survivors.

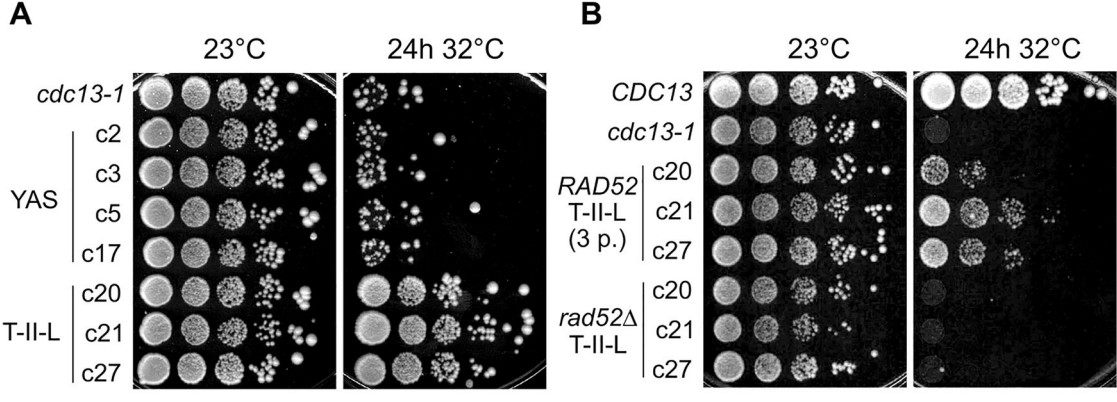

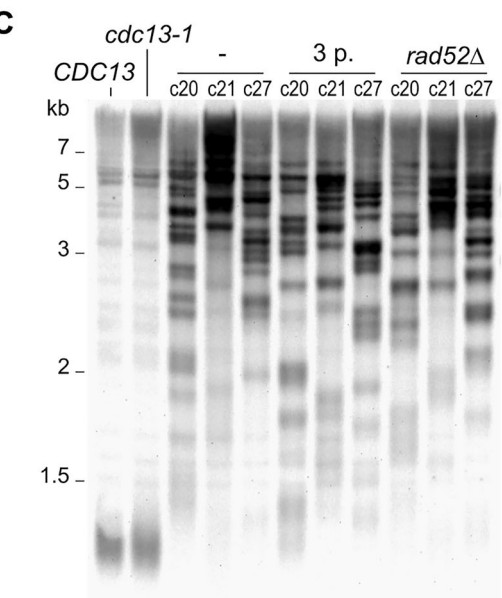

**Figure 6. T-II-L survivors show a *RAD52*-dependent ability to resist second uncapping.**

(A) Spot assay showing survival and growth either at a constant 23 °C (left) or after a second 24-h telomere uncapping (right) for the indicated strains. (B) Same as (A) for the indicated strains. "*RAD52* T-II-L (3 p.)" c20, c21, and c27 correspond to the same strains as in (A) but were passaged three more times. (C) TRF Southern blots of the indicated strains. The type-II-like clones c20, c21, and c27 were analyzed once they were isolated and after three additional passages ("3 p."). *RAD52* was deleted in the same clones, and the transformation and selection procedure required the equivalent of three passages. See Fig. EV5C for reordered lanes grouped by clone. Source data are available online for this figure.

To assess whether recombination contributed to this observed resistance, we deleted *RAD52* in the 3 previously analyzed type-II-like survivor strains. As we observed that long telomeres in type-II-like strains shortened over successive passages (Fig. EV5B), we accounted for the cell divisions occurring during the transformation procedure to delete *RAD52* by passaging the control *RAD52* parents three times. Strikingly, the *rad52Δ* strains completely lost resistance to telomere uncapping (Fig. 6B), indicating that recombination was required for this protective effect. Importantly, this loss was not attributable to a reduction in telomere length, as TRF Southern blotting showed that the *rad52Δ* mutants retained long telomeres comparable to those of their parental strains (Figs. 6C and EV5C).

Notably, the *RAD52* type-II-like strains shown in Fig. 6B exhibited reduced survival compared to those in Fig. 6A, which was

particularly apparent for clone c20. Since the former were passaged three additional times, we attributed this decreased viability to telomere shortening that occurred during the 50–75 divisions, including potentially sporadic events of rapid telomere deletion.

Collectively, these findings demonstrate that, together, Rad52 and long telomeres protect against a second uncapping.

## Discussion

How the genome is affected by telomere uncapping remains an outstanding question in the telomere field. Long-term survivors of permanent telomere uncapping have been previously selected after many passages and in the context of mutated checkpoint or DNA processing pathways, thus offering little information regarding the

immediate molecular and cellular response to telomere uncapping in checkpoint-proficient conditions. Using a transient uncapping experimental scheme combined with molecular methods and long-read nanopore sequencing, we provide a comprehensive character-ization of the genome rearrangements following telomere uncap-ping, their genetic requirements and their potential functional role in protecting against further telomere deprotection.

## Early onset of genomic instability, even with a functional checkpoint

Our experimental scheme allowed us to show that genome rearrangement can arise as early as during the first cell cycle after telomere uncapping, as suggested by the presence of sharply shifted PFGE bands (Fig. 2A), indicating rearrangements present in the whole clonal population. Genomic heterogeneity in *cdc13-1 cdc5-ad* survivor subclones indicates that genome instability persists for at least two cell divisions after uncapping is relieved and cells resume their cell cycle. We thus propose that transient telomere uncapping and ssDNA accumulation initiate a cascade of recombination events, with the first rearrangements arising within a few hours and others unfolding during the recovery phase in the progeny. We note that, while it is well established that Exo1-dependent resection is the major contributor to the accumulation of ssDNA at uncapped telomeres (Garvik et al, 1995; Maringele and Lydall, 2002; Zubko et al, 2004), consistent with the increased survival in the *cdc13-1 exo1Δ* mutant in our assay (Fig. EV3A), a defect in the CST's function in telomere replication through the interaction with Polα-primase and C-strand fill-in (Grossi et al, 2004; Qi and Zakian, 2000) might also contribute to the extent of ssDNA and to the subsequent rearrangements. Importantly, rearrangements occurred in checkpoint-proficient cells, suggesting that they did not result from uncontrolled cell cycling with broken DNA, mis-segregated chromosomes, or other abnormal structures, but rather from a canonical response to ssDNA exposed at uncapped telomeres. The use of a checkpoint-proficient background in our study stands in contrast with previous works where Cdc13-independent survivors and telomere elongation could be observed after extensive incubation at restrictive temperatures only if checkpoint or DNA processing pathways are simultaneously mutated (Grandin et al, 2001; Grandin and Charbonneau, 2003, 2013; Zubko and Lydall, 2006).

Therefore, transient telomere deprotection alone can be a source of genome instability, a finding with important ramifications for understanding the potential contribution of partial and transient telomere deprotection during the initial steps of tumorigenesis when checkpoints are still functional.

## Mechanisms underlying telomere-uncapping-driven rearrangements

Since transient-telomere-uncapping-driven telomeric rearrange-ments are dependent on *RAD52* and *POL32*, we propose that homologous recombination and BIR are the main molecular mechanisms underlying their formation. Additionally, both Rad51 and, to a lesser degree, Rad59 are important for Y'-associated rearrangements, especially amplifications. However, the Rad51 requirement is partial, since tandem amplification of up to 8 Y' elements could still be observed in *cdc13-1 rad51Δ* survivors,

which stands in contrast to the strictly Rad51-dependent Y' amplification observed in telomerase-negative type I survivors (Teng et al, 2000; Chen et al, 2001). Interestingly, the fraction of type-II-like survivors measured in the *cdc13-1 rad51Δ* strain is not greater than in the *cdc13-1* strain —if anything, they tend to be less frequent in the absence of *RAD51*—, another major difference compared to telomerase-negative type II survivors, whose relative proportion is strongly increased in the absence of *RAD51* (Teng et al, 2000). We found no case of type-II-like telomere elongation in *cdc13-1 rad59Δ* survivors, suggesting that, as for telomerase-negative survivors, Rad59 might promote Rad52-mediated anneal-ing and recombination between TG$_{1-3}$ sequences (Churikov et al, 2014; Kockler et al, 2021), possibly through a Rad51-independent BIR pathway that relies on microhomologies (Ira and Haber, 2002). The additive effect of deleting *RAD51* and *RAD59* indicates that they act on independent pathways, at least partially.

With respect to telomere elongation in type-II-like survivors, our detailed sequence analyses strongly suggest the use of t-circles as templates for rolling circle amplification, resulting in perfect tandem repeats of specific motifs of 50–600 bp. Overall, t-circle-mediated telomere elongation accounts for up to ~45% of all telomere sequences in type-II-like survivors and most likely more. We could trace the sequence of some tandem repeats back to a unique telomere sequence in the corresponding control strains, strongly suggesting that t-circles might emerge through excision of an endogenous telomere sequence. Interestingly, t-circles were also detected in telomerase-negative type II survivors in previous studies (Aguilera et al, 2022; Larrivée and Wellinger, 2006), in which they were proposed to be the product of telomeric recombination or replication fork stalling at telomere sequences. Similar mechanisms could be at play in telomere-uncapping-induced type-II-like survivor and would activate regardless of the initial triggering signal.

Y'-associated survivors represent the most frequent survivor outcome of transient telomere uncapping, and we propose that extensive resection following telomere uncapping favors Y' rearrangements owing to the homology length these elements provide and to the presence of internal telomeric sequences, as well as their organization as tandem repeats at some subtelomeres. When the homology used is within the Y' element itself, new hybrid Y' elements can emerge, suggesting that this process might participate in Y' element evolution and diversification. Telomere elongation would occur less frequently and later after telomere uncapping, but not always, since we observed rearrangement patterns involving >1-kb-long telomere sequences interspersed between Y' elements (Fig. 3D; Dataset EV2), suggesting cases where telomere elongation precedes the invasion into an interstitial telomeric sequence and subsequent Y' rearrangements.

Overall, extensively resected uncapped telomeres engage in multiple homology-dependent mechanisms, resulting in Y'-asso-ciated rearrangements, t-circle mediated telomere elongation, and even more distal subtelomere duplication and translocation. While some similarities can be drawn with the mechanisms and the genetic requirements of post-senescence telomerase-independent survivors, significant differences in the response to telomere uncapping and critically short telomeres are also observed and might be due to differences in the initial length and state of the telomeres, their processing, and the signaling pathways they activate. Additionally, during and after uncapping, telomerase is

still present in the cell and might play an important role in preserving or replenishing the telomeric 3' end sequence if needed.

## Survival to transient telomere uncapping is partially dependent on homologous recombination

Loss of viability is observed when telomeres are uncapped for more than 6 h. While the exact causes of cell death remain unclear, survival partially depends on recombination factors and is associated with subtelomere and telomere rearrangements. However, altered Y' element organization in Y'-associated survivors, the most frequent rearrangement pattern, does not confer any selective advantage in response to a second round of transient telomere uncapping. We thus speculate that recombination factors might directly promote survival, at least for uncapping of 24 h, by engaging molecularly with chromosome extremities, regardless of the outcome of the molecular transaction. For example, by directly interacting with ssDNA, Rad52, Rad51, and Rad59 might stabilize the end structure and limit excessive checkpoint activation, thereby acting as a protective cap for the telomeres.

On the other hand, we demonstrate that the massively elongated telomeres in type-II-like survivors strongly promote survival to a second telomere uncapping event. This finding is consistent with previous studies that used permanent telomere uncapping protocols, often together with additional mutations in checkpoint or resection pathways (Grandin et al, 2001; Grandin and Charbonneau, 2003; Zubko and Lydall, 2006), where survivor cells systematically displayed very long and heterogeneous telomeres. However, we show that an active *RAD52*-dependent homologous recombination pathway is required for type-II-like cells to survive a second transient telomere uncapping. A parallel can be drawn with observations that deletion of *RAD52* in type II telomerase-negative survivors very quickly leads to loss of viability, despite the presence of long telomeres (Teng and Zakian, 1999). This finding is consistent with our model, which proposes that the presence of recombination factors at telomeres or their activity is important for promoting survival to telomere uncapping. In type-II-like survivors, as compared to the parental strain, the very long telomeres would provide much more homology for Rad52-dependent annealing, thus promoting survival. Conceptually, we thus propose that recombination factors or the molecular structures formed by long telomeres undergoing recombination serve as an alternative cap for telomeres when Cdc13 is dysfunctional.

## Methods

### Reagents and tools table

| Reagent/resource | Reference or source | Identifier or catalog number |
|---|---|---|
| **Experimental models** | | |
| *Saccharomyces cerevisiae* W303 background | (Thomas and Rothstein, 1989) | N/A |
| **Recombinant DNA** | | |
| N/A | | |
| **Antibodies** | | |

| Reagent/resource | Reference or source | Identifier or catalog number |
|---|---|---|
| N/A | | |
| **Oligonucleotides and other sequence-based reagents** | | |
| Telomere-specific oligonucleotide probe (5'-CCCACCACACACACCCACACCC-3') biotinylated at both ends | Eurofins Genomics | N/A |
| **Chemicals, enzymes and other reagents** | | |
| YPD broth | Sigma-Aldrich | Y1375-1Kg |
| YPD agar | Sigma-Aldrich | A1296-500G |
| QIAGEN Genomic-tip 100/G | Qiagen | 10243 |
| Zymolyase 20 T | Euromedex | UZ1000 |
| RNAse A | Euromedex | 9707 |
| SRE Kit | PacBio | 102-208-300 |
| Adapter ligation kit | Oxford Nanopore Technologies | SQK-LSK109 and SQK-LSK110 |
| Native barcodes | Oxford Nanopore Technologies | EXP-NBD104 |
| Nanopore flowcell R9.4 | Oxford Nanopore Technologies | FLO-MIN106D |
| Nanopore flowcell R10.4 | Oxford Nanopore Technologies | FLO-PRO114M |
| Wash kit | Oxford Nanopore Technologies | WSH004 |
| NEBnext companion module for Oxford Nanopore Technologies | New England Biolabs | E7180S |
| Blunt/TA ligase master mix | New England Biolabs | M0367L |
| SeaKem GTG Agarose | Lonza | 50070 |
| Phenol:chloroform:isoamyl (25:24:1) | Sigma-Aldrich | P3803 |
| XhoI FastDigest | Thermo Fisher Scientific | FD0694 |
| Gel Loading Dye, Purple (6X) | New England Biolabs | B7024S |
| Hybond-XL | GE Healthcare | RPN 203 S |
| Nucleic Acid Detection Blocking Buffer | Thermo Fisher Scientific | 89880 A |
| Streptavidin, Alkaline Phosphatase Conjugate | Invitrogen/Thermo Fisher Scientific | S921 |
| CDP-*Star*™ Substrate | Invitrogen/Thermo Fisher Scientific | T2146 |
| **Software** | | |
| Guppy v6.1.5 | https://github.com/nanoporetech/pyguppyclient | N/A |
| Dorado v1.3.0 | https://github.com/nanoporetech/dorado | N/A |
| Filtlong 0.2.0 | https://github.com/rrwick/Filtlong | N/A |
| Canu 2.2 | (Koren et al, 2017) | N/A |
| Flye 2.9.1 | (Kolmogorov et al, 2019) | N/A |
| Racon 1.5.0 | (Vaser et al, 2017) | N/A |

| Reagent/resource | Reference or source | Identifier or catalog number |
|---|---|---|
| Medaka 1.11.3 or 2.0.1 | https://github.com/nanoporetech/medaka | N/A |
| Telofinder | (O'Donnell et al, 2023) | N/A |
| LRSDAY 1.7.2 | (Yue and Liti, 2018) | N/A |
| Clair3 v1.0.10 | (Zheng et al, 2022) | N/A |
| Sniffles2 | (Smolka et al, 2024) | N/A |
| PAV | (Ebert et al, 2021) | N/A |
| Other | | |
| MK1C Minion sequencer | Oxford Nanopore Technologies | MIN-101C |
| PromethION 2 Solo sequencer | Oxford Nanopore Technologies | PRO-SEQ002 |
| CHEF-DR III System | Bio-Rad | 962BR1898 |

## Yeast strains

All strains are from the W303 background (*ura3-1 trp1-1 leu23,112 his3-11,15 can1-100*) (Thomas and Rothstein, 1989) corrected for *RAD5* and *ADE2* (Table EV1). Most strains carry the *cdc13-1* allele and were grown routinely at the permissive temperature of 23 °C in YPD (yeast extract, peptone, and dextrose) media. Deletion strains were created using PCR-based methods as described in Longtine et al, 1998. Point mutations were introduced using Cas9-mediated gene targeting as described in Anand et al, 2017.

## Transient uncapping survival assay

Cells were inoculated in YPD and grown overnight at 23 °C. The culture was then diluted and grown in exponential phase until it reached a concentration of $1–2 \times 10^7$ cells/mL. Between $5 \times 10^5$ and $5 \times 10^6$ cells were plated on solid YPD media preheated at 32 °C and then kept at 32 °C for 3 to 24 h before the temperature was shifted back to 23 °C until colony formation (~3 days). Another plate was inoculated with ~500 cells of the same liquid culture and kept at 23 °C to measure plating efficiency and to be used for normalization. The number of cells plated was dependent on the expected survival frequency of each strain and was chosen in order to yield between 50 and 300 individual colonies. The colonies on the plate that was placed transiently at 32 °C and on the plate kept at 23 °C were then counted to calculate the survival frequency. Colonies randomly selected for further investigation were subsequently subcloned and then grown at 23 °C.

## Genomic DNA extraction

Two genomic DNA extraction protocols were used in this study for subsequent nanopore sequencing. For the first protocol, QIAGEN Genomic-tip 100/G was used following the manufacturer's protocol and starting with a total of $5 \times 10^9$ cells from an overnight liquid culture. For the second protocol, $1–2 \times 10^9$ cells from an overnight liquid culture were pelleted, washed with 500 μL of spheroblasting solution (1 M sorbitol, 50 mM KPO$_4$, and 10 mM EDTA),

resuspended in 500 μL spheroblasting solution supplemented with 5 μL of β-mercaptoethanol and 1.25 mg of zymolyase 20 T and incubated for 30 min at 37 °C. Spheroblasts were pelleted at 2500×g for 3 min, resuspended in 500 μL of lysis solution (0.1 M Tris HCL at pH 8.0, 50 mM EDTA, 0.5 M NaCl, 1% PVP40, and 2.5% SDS) with 1.5 mg of RNase A, and then incubated for 45 min at 50 °C. 1 mL of TE and 500 μL of 5 M potassium acetate were added and gently mixed before two centrifugation steps of 10 min at 9500×g at 4 °C to clarify the supernatant. DNA was precipitated from the supernatant with an equal volume of isopropanol and washed with 70% cold ethanol before final elution in 40 μL of H$_2$0.

## Library preparation and sequencing

The sequencing libraries were prepared following Oxford Nanopore Technologies (ONT) protocols for genomic DNA without pre-amplification and using LSK109 or LSK110 kits (Table EV2). Genomic DNA was enriched for long fragments using the SRE kit (PacBio). DNA libraries were sequenced on R9.4 or R10.4 Nanopore flow cells in either a MinION Mk1C or PromethION P2 sequencer, respectively, with default parameters on the MinKNOW operating software. For each run, 500 ng of DNA was loaded and sequenced for 8–24 h. In case of DNA reloading, the flow cell was washed using ONT's wash kit following the instructions.

## Pulsed field gel electrophoresis

Agarose plugs containing $2 \times 10^8$ yeast cells were prepared as described in (Török et al, 1993) and sealed in a 0.5X TBE (0.89 M Tris, 0.89 M boric acid, and 20 mM EDTA) 1% Seakem GTC agarose gel. PFGE was run on the CHEF-DRII system (Bio-Rad) with the following program: 6 V/cm for 10 h with a switching time of 60 s, followed by 6 V/cm for 17 h with a switching time of 90 s. The reorientation angle was set to 120° throughout the entire run.

## TRF Southern blot

Genomic DNA was extracted from cultures using a standard phenol:chloroform:isoamyl (25:24:1) purification procedure and isopropanol precipitation. A sample of 2 μg of genomic DNA was digested with XhoI, and the products were ethanol-precipitated, resuspended in loading buffer (purple gel loading dye 6X, New England Biolabs), and resolved on a 1.1% agarose gel for 6 h at 3.6 V/cm. The gel was then soaked in a denaturation bath (0.4 M NaOH, 1 M NaCl) for 20 min and transferred by capillary action to a charged nylon membrane (Hybond XL, GE Healthcare). A telomere-specific oligonucleotide probe (5′-CCCACCACACA-CACCCACACCC-3′) biotinylated at both ends was used for hybridization. After hybridization of the probe, the membrane was washed 3 × 5 min in wash buffer (58 mM Na2HPO$_4$, 17 mM NaH$_2$PO$_4$, 68 mM NaCl, and 0.1% SDS). The membrane was next processed for detection with three successive incubations (5, 5, and 30 min) in blocking buffer (Thermo Scientific, Nucleic Acid Detection Blocking Buffer) before a 30 min incubation with alkaline phosphatase-conjugated streptavidin (Invitrogen) diluted in blocking buffer (0.4 g/mL). The membrane was then washed again 3 × 5 min in wash buffer, incubated for 2 × 2 min in assay buffer (0.1 M Tris, 0.1 M NaCl, pH 9.5) and for 5 min in CDP-Star

substrate (Invitrogen) before being imaged with a Bio-Rad ChemiDoc Touch device by chemiluminescence.

## Whole genome assemblies

Data from sequencing was basecalled using Guppy with model dna_r9.4.1_450bps_sup.cfg for R9 data and Dorado with model dna_r10.4.1_e8.2_400bps_sup@v5.0.0 for R10 data. Reads were filtered using Filtlong to keep only the best 500–700 Mb using the filtering parameters --length_weight 6 --mean_q_weight 3 --window_q_weight 1 –min_length 1000. The genomes were assembled using Canu 2.2 (Koren et al, 2017) and Flye 2.9.1 (Kolmogorov et al, 2019), using standard parameters for Canu and the option --nano-hq for Flye, then polished using one round of Racon v1.5.0 (Vaser et al, 2017) followed by two rounds of Medaka (v1.11.3 for R9 data, v2.0.1 for R10 data). The output assemblies were mapped against the S288C reference genome for contig name annotation, and short contigs (<70 kb) and contigs corresponding to mitochondrial DNA were removed. To obtain the final assembly, reads were mapped against Canu and Flye assemblies, telomeric regions were manually inspected using Tablet, and the best contigs were retained for each chromosome. Trimming of contigs was manually performed when overassembly at telomeres was detected through loss of coverage. The statistics of each sample for the reads and assemblies can be found in Table EV2.

## Annotation of telomeric regions

Telomeres were detected on assemblies using Telofinder (O'Donnell et al, 2023) (https://github.com/GillesFischerSorbonne/telofinder). X elements were detected in the assemblies using LRSDAY 1.7.2 (Yue and Liti, 2018). Y' elements were detected by aligning known Y' elements from the reference genome against the considered assembly using BLAST 2.12.0, and then taking the union of all alignments whose lengths were greater than 200 bp.

## Detection of gene variants in survivors

Gene annotation of the control genome assembly was obtained using LRSDAY 1.7.2 (Yue and Liti, 2018). Variant calling in the genomes of survivor strains was carried out using Clair3 v1.0.10 (Zheng et al, 2022). Variants identified when mapping control reads to the control assembly were considered assembly-derived artifacts and were therefore excluded from downstream analyses. In addition, variants with a quality score below 10 were filtered out. Genes harboring variants in at least two independent survivor strains were classified as candidate targets of selection during telomere uncapping. ORFs corresponding to Y' elements were excluded from this analysis. The resulting gene set was subsequently compared with datasets from three independent genetic screens performed in *cdc13-1* strains (Addinall et al, 2008, 2011; Holstein et al, 2017).

## Y' element labeling and clustering

Because nanopore sequencing makes most errors in homopolymers, the detected Y' sequences were transformed by condensing all homopolymers of length >4 into homopolymers of length 4. Doing so increased robustness and yielded a 100% similarity between the Y' elements of the two independent assemblies of our control strain. The similarity measure was derived from BLAST alignment results, using the best alignment between each pair of sequences. The formula used was sim = %identity × alignment_length/alignement_span, where alignment_span is the length of the alignment plus the length of the unaligned flanking regions, and %identity was obtained from BLAST.

To assess the precision of our method, another set of reads was split and assembled into two independent assemblies, which yielded similarity values between homologous Y' between 99.985 and 100%. Labels were then assigned to the Y' elements of our control strain. Two or more Y' elements were given the same label if their similarity score was higher than 99.9%.

Hierarchical clustering of the Y' elements of our control strain was performed on their similarity matrix using Seaborn's clustermap function with default parameters. The optimal number of 11 clusters was determined by determining the elbow point of the silhouette score.

## Subtelomeric alteration detection

To detect Y' element structure alterations, we compared the labels assigned to each element to the labels of the elements in the corresponding control strain. Any label that was present at a position in the control and not in a survivor was considered a loss. Any label present at a position in a survivor and not in the control was considered a gain.

To detect subtelomeric events not including Y' elements or telomeres, sequences upstream of the X element at each extremity were aligned over 15 kb using BLAST against their control counterparts, and any deviation from perfect alignment was flagged for manual investigation.

To check whether a new Y' sequence not present in the control genome could be explained by a mosaic of elements in the control genome, we used a custom Python script to look for perfect alignments of size >50 between a new Y' element and all known Y' elements. A Y' element was considered mosaic if it could be covered by the union of such alignments.

## T-circle signature detection in T-II-L telomeres

Telomeres from all T-II-L assembled genomes, as well as the control genome, were scanned for perfect tandem repeats with repeat units of 30 bp or longer. As the control genome contained tandems with repeats of up to 45 bp, repeats smaller than 50 bp were ignored for all the following steps. Repeats occurring in arrays of at least three units, and therefore unlikely to result from a simple duplication based on other mechanisms, were classified as t-circle sequences. When several tandems were composed of identical or highly similar repeats (alignment score greater than 0.98), only the most common sequence was selected, and circles were labeled in the figures according to the size of this sequence. Each t-circle sequence was then matched against a sliding window of equal size, across telomeres of the same survivor as well as telomeres of other survivors and the control strain. Alignments were done using Python's Bio.Align.PairwiseAligner with parameters mode = "local", match_score = 1, mismatch_score = −1, open_gap_score = −1, extend_gap_score = −0.9, and the circular nature of the sequence was accounted for by duplicating the t-circle sequence before alignment, thus providing all circular permutations of the

sequence. Matches were defined when the normalized score was greater than 0.98. T-circle repeats that were detected in another assembly in more than one copy (usually relatively small sequences) were removed from analysis, as it meant the sequence was not specific enough. If the t-circle repeats were detected in the assembly of the corresponding control strain in exactly one copy, we considered the latter as being the locus of origin of the t-circle sequence.

## Statistical analyses

For all statistical analyses, at least three independent experiments were performed. No randomization and no blinding were needed in any experiment. No sample was excluded from analysis. Statistical tests were performed using the Python library Scipy 1.15.0.

## Data availability

All new sequencing data were deposited in the European Nucleotide Archive (https://www.ebi.ac.uk/ena/browser/home) under the project PRJEB93986. The reads data (FASTQ files) can be found here: https://www.ebi.ac.uk/ena/browser/view/PRJEB93986?show=reads. The assembly data (FASTA files) can be found here: https://www.ebi.ac.uk/ena/browser/view/PRJEB93986?show=wgs. All materials and strains generated in this work are available upon request.

The source data of this paper are collected in the following database record: biostudies:S-SCDT-10_1038-S44319-026-00717-4.

## Peer review information

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

## Acknowledgements

We thank Stéphane Delmas and Teresa Teixeira for their critical reviews of the paper. We thank Nicolas Agier for his help and advice with Nanopore sequencing and subsequent analyses. Work in ZX's lab was supported by Ville de Paris (Program Émergence(s)), the Emergence grant of Sorbonne Université, Ligue Contre le Cancer (Subvention Recherche Scientifique 2022), Fondation ARC pour la recherche sur le cancer (ARCPJA202160003865 and ARCPGA2023110007341_7967), and by Agence Nationale de la Recherche (ANR-24-CE12-7740-01 and ANR-24-CE12-0083-03).

## Author contributions

**Liébaut Dudragne**: Conceptualization; Formal analysis; Investigation; Visualization; Methodology; Writing—original draft; Writing—review and

editing. **Clotilde Garrido**: Investigation; Methodology; Writing—review and editing. **Oana Ilioaia**: Investigation; Methodology; Writing—review and editing. **Juliana Silva Bernardes**: Conceptualization; Formal analysis; Supervision; Methodology; Writing—original draft; Writing—review and editing. **Zhou Xu**: Conceptualization; Supervision; Funding acquisition; Validation; Writing—original draft; Project administration; Writing—review and editing.

Source data underlying figure panels in this paper may have individual authorship assigned. Where available, figure panel/source data authorship is listed in the following database record: biostudies:S-SCDT-10_1038-S44319-026-00717-4.

## Disclosure and competing interests statement

The authors declare no competing interests.

# Expanded View Figures

**Figure EV1. Heterogeneous genome rearrangements at telomeres and subtelomeres.**

(A) TRF Southern blot analysis of 16 *cdc13-1* survivor clones in addition to those shown in Fig. 2B. The control lane for *cdc13-1* 23 °C is taken from a different Southern blot. (B) Southern blot performed on the same membrane as in Fig. 1B, hybridized with a Y'-specific probe. The bottom panel shows ethidium bromide staining of the gel showing total DNA loading prior to transfer. (C) PFGE of four subclones of clones c2, c3 (also shown in Fig. 2C), and c5. The orange boxes indicate bands with distinct migration patterns in the four subclones. (D) PFGE of 7 *cdc13-1 cdc5-ad* transient uncapping survivor clones, as well as a *cdc13-1* control strain grown constantly at 23 °C (same control as in Fig. 2A). Compared to the control strain, six out of seven survivor clones exhibited apparent chromosome size shifts, marked by orange arrows. (E) Same as (C) in a *cdc13-1 cdc5-ad* strain.

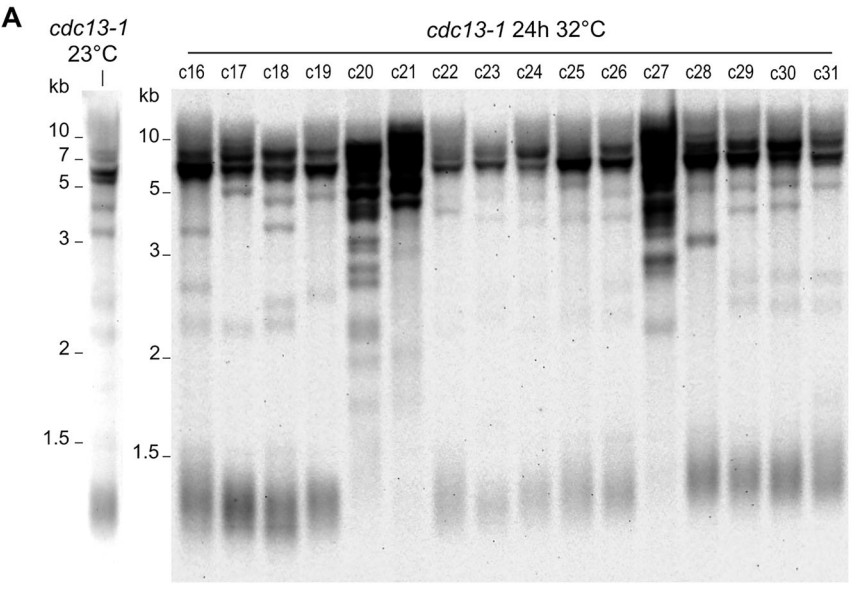

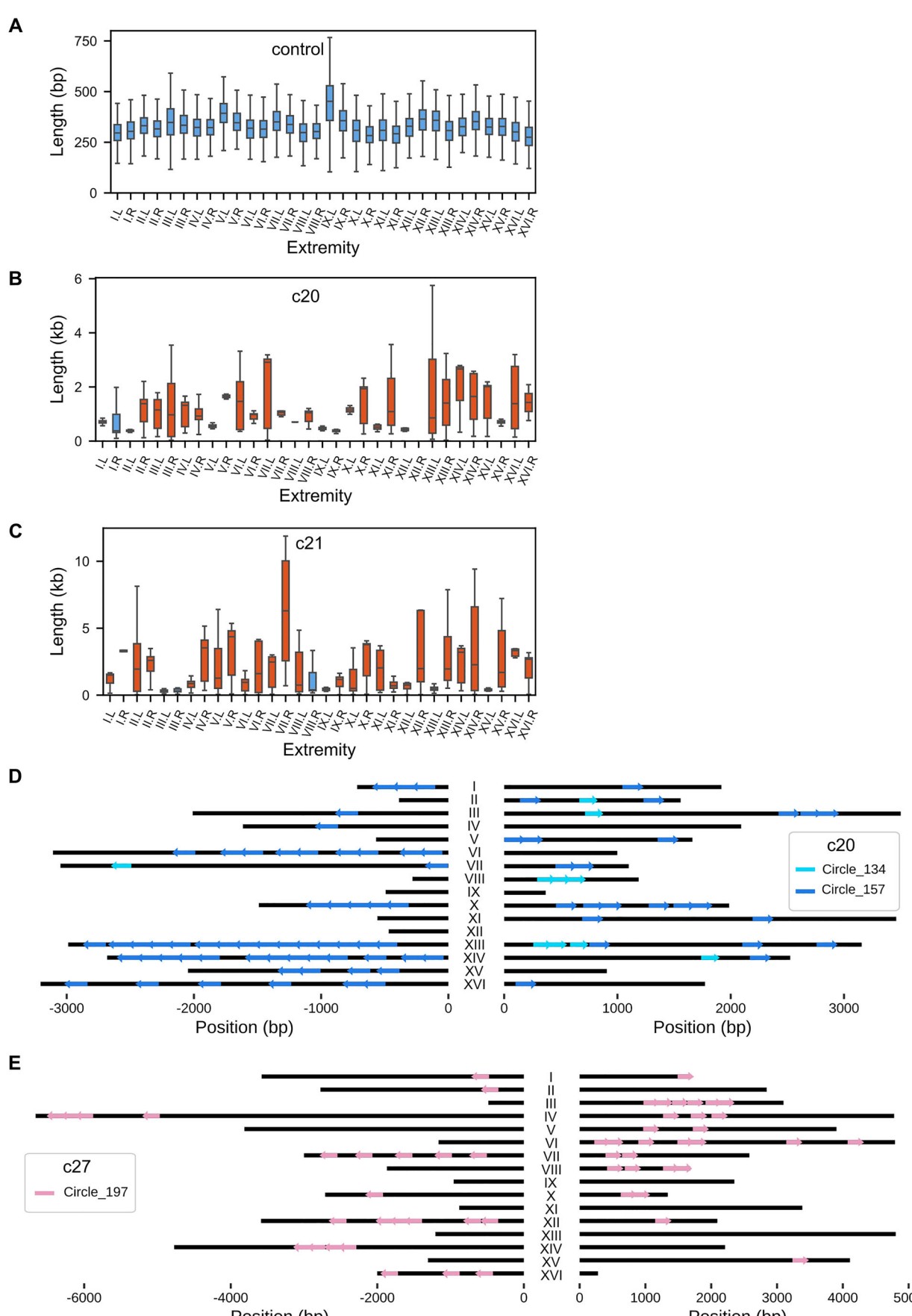

**Figure EV2.  Telomeres are elongated heterogeneously in T-II-L survivors and contain large perfect tandem repeats.**

(**A**) Boxplots representing the telomere length distribution at each chromosome extremity of the control *cdc13-1* strain grown at 23 °C. Same data as in Fig. 4C left with a different scale on the y-axis. The boxplots follow the same specifications as in Fig. 4C. (**B**, **C**) Same as (**A**) for type-II-like survivor clones c20 and c21. The blue boxplots correspond to distributions with a mean telomere length <500 bp. The boxplots follow the same specifications as in Fig. 4C. The number of reads used to draw the boxplots are comprised between $n = 3$ and $n = 31$ for (**B**), except for XII.R and VIII.L, which have $n = 0$ (no boxplot) and $n = 1$ (single line), respectively, and are comprised between $n = 4$ and $n = 32$ for (**C**). (**D**, **E**) Visualization of putative circle-originating sequences detected in the left and right telomeres of the 16 chromosomes from clones c20 and c27. The circle number corresponds to the length of the circle sequence. Black horizontal lines represent the assembled telomeric sequences.

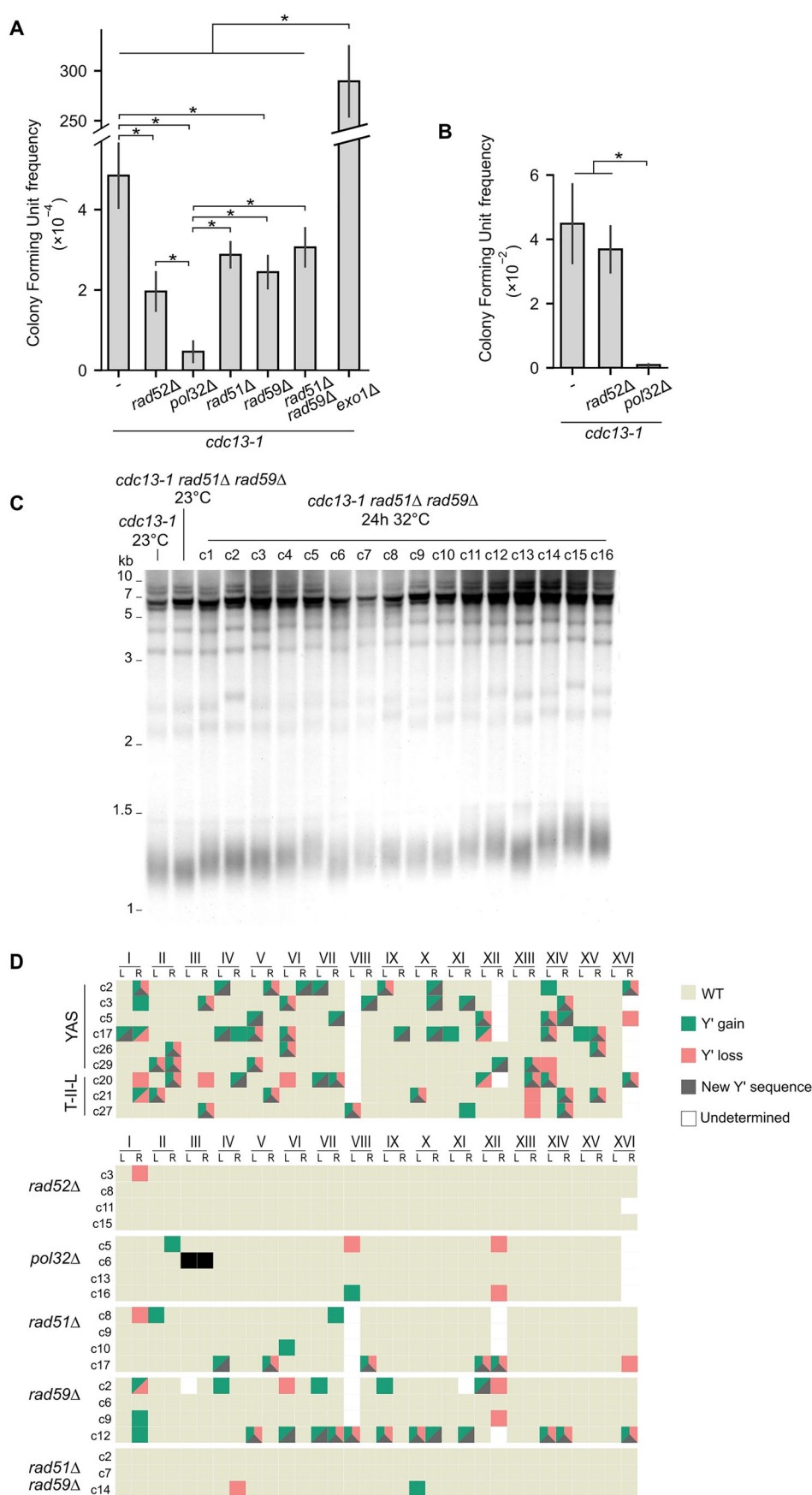

Figure EV3.  Survival and alterations in single and double mutants in the *cdc13-1* background.

(A) Colony formation frequency after 24 h of transient telomere uncapping and return to 23 °C in the indicated mutants. A control plate with the *cdc13-1* strain was kept only at 23 °C for normalization. The number of independent samples is $n = 7, 4, 4, 4, 5, 3$, and 3, following the order of the x-axis. The error bars correspond to the standard error of the mean. Asterisks indicate statistically significant differences (Student's *t*-test: for *exo1Δ* against the other strains from left to right, *p* value $= 9.0 \times 10^{-7}$, $2.0 \times 10^{-4}$, $1.9 \times 10^{-4}$, $2.0 \times 10^{-4}$, $3.2 \times 10^{-5}$, and $1.3 \times 10^{-3}$; for other asterisks, from top to bottom, *p* value = 0.040, 0.032, 0.0032, 0.0036, 0.0062, 0.0011, and 0.034). (B) Same as (A) for the indicated strains, but following 12 h of transient telomere uncapping, with $n = 4, 6$, and 5, following the order of the x-axis. Student's *t*-test, ∗: *p* value = 0.0048 for WT against *pol32Δ* and *p* value = 0.0015 for *rad52Δ* against *pol32Δ*. (C) TRF Southern blots of *cdc13-1 rad51Δ rad59Δ* survivor colonies. A culture from the same strain that was maintained at 23 °C was used as a control. (D) Map of chromosome extremity alterations further detailing modified Y' structures from Figs. 3B and 5E. In this map, a substitution is simultaneously a loss and a gain of the Y' element.

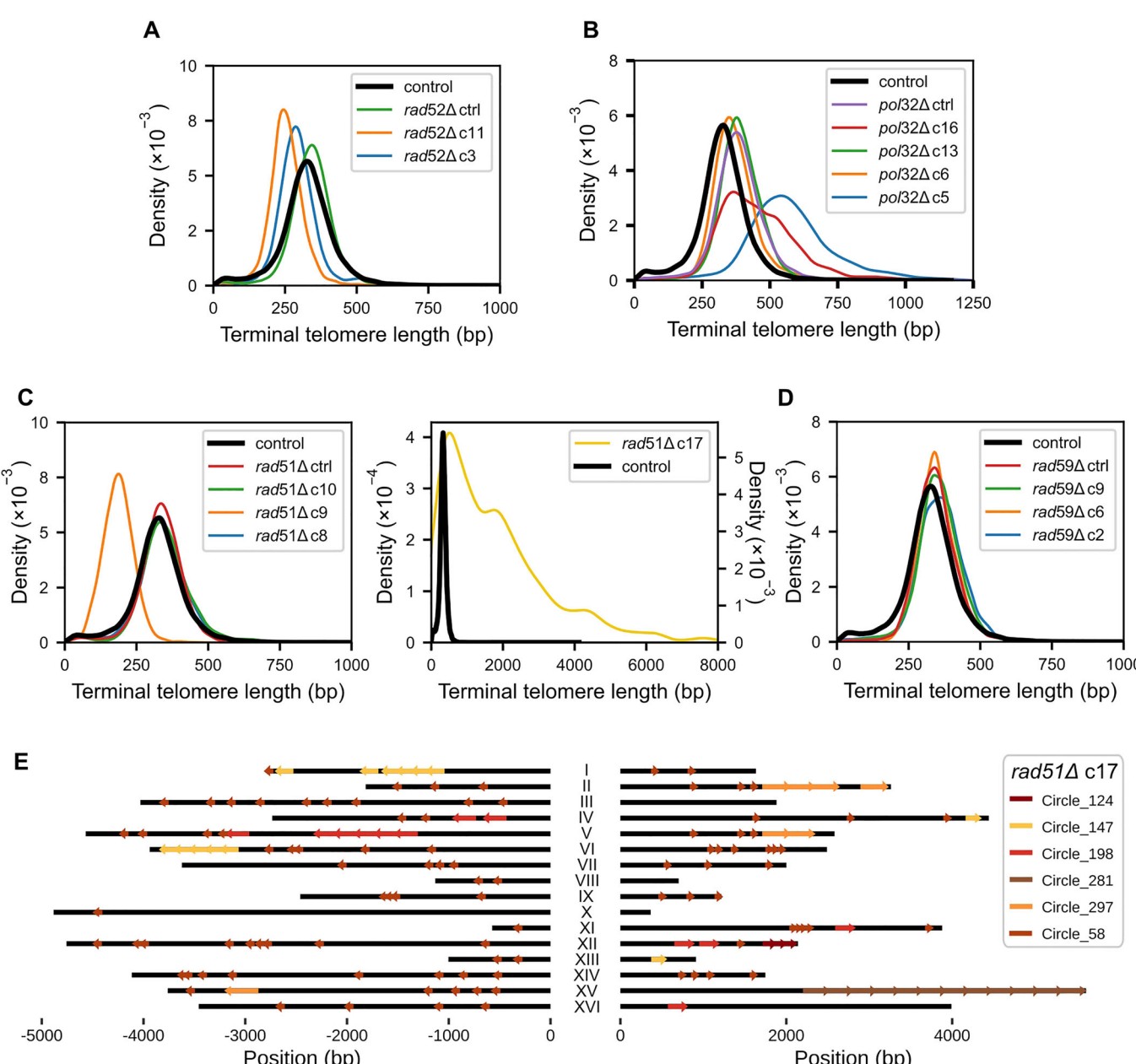

**Figure EV4. Telomere length distributions of transient uncapping survivors in the *rad52Δ*, *pol32Δ*, *rad51Δ*, and *rad59Δ* mutants.**

(A–D) Telomere length distributions of the indicated strains derived from nanopore sequencing reads, compared to the control (black line). In (C), the left plot displays the telomere length distribution of *rad51Δ* YAS survivors, while the right plot displays the telomere length distribution of the *rad51Δ* type-II-like survivor (c17), with a different scale for the x-axis. (E) Visualization of putative circle-originating sequences detected in the left and right telomeres of the 16 chromosomes of the *rad51Δ* T-II-L clone c17. The circle number corresponds to the length of the circle sequence. Black horizontal lines represent the assembled telomeric sequences.

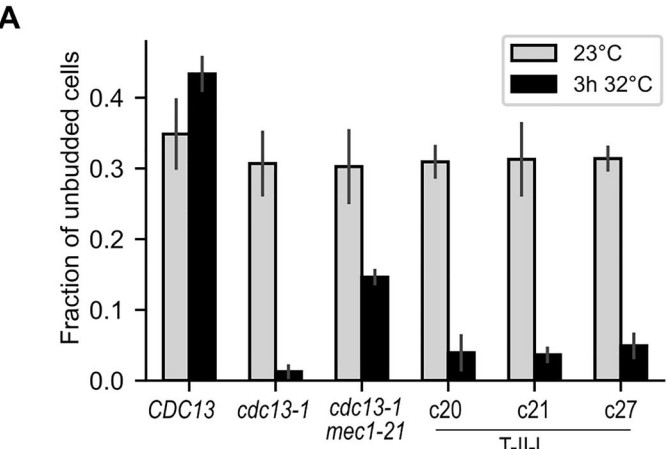

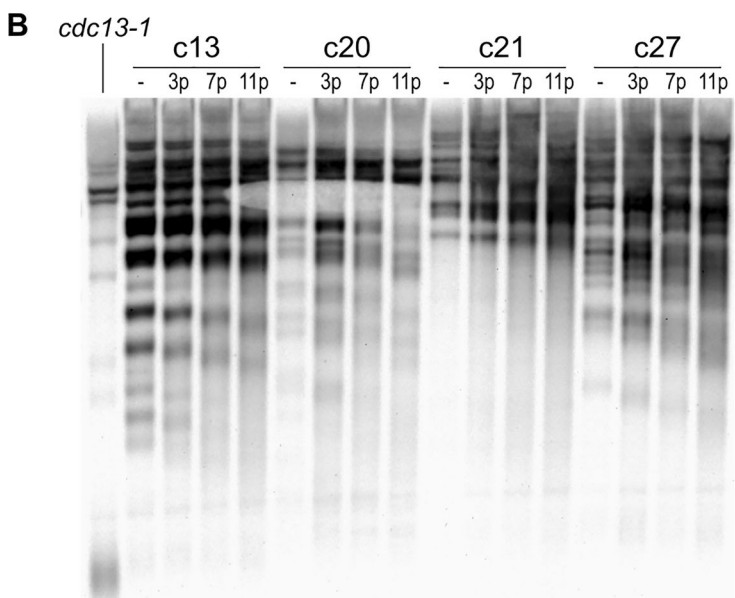

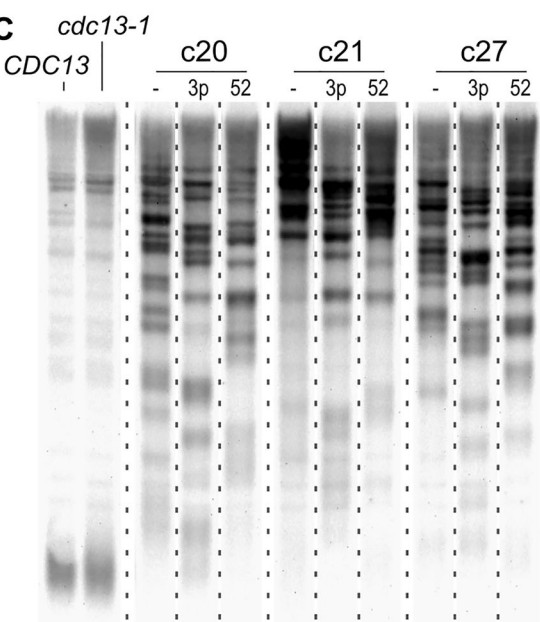

**Figure EV5. Behavior of type-II-like survivors in response to a second telomere uncapping and through further passages.**

(A) Fraction of unbudded cells in exponentially growing cultures at 23 °C or after 3 h at 32 °C. Unbudded cells are more abundant in the partially checkpoint-deficient mutant *mec1-21 cdc13-1* than in the control *cdc13-1* after 3 h at 32 °C. *n* = 3 independent cultures for each condition. The error bars correspond to the standard error of the mean. (B) TRF Southern blot of 4 type-II-like clones, passaged for an additional 3, 7, and 11 passages after initial subcloning. Contrary to Fig. 6C, restreaks were performed on bulk patches of cells instead of colonies. (C) Figure identical to Fig. 6C, except that the lanes have been digitally reordered to facilitate visual interpretation per clone. "52" indicates the *rad52Δ* derivative of the indicated strain.

