## [Peer Review File · EMBO Reports]

Transient telomere uncapping triggers telomeric and subtelomeric rearrangements

Liébaud Dudragne, Clotilde Garrido, Oana Ilioaia, Juliana Bernardes, and Zhou Xu

Corresponding author(s): Zhou Xu (zhou.xu@polytechnique.org) , Juliana Bernardes (jusilvabernardes@sb-roscoff.fr)

Review Timeline:

Submission Date:	15th Aug 25
Editorial Decision:	10th Oct 25
Revision Received:	23rd Dec 25
Editorial Decision:	13th Jan 26
Revision Received:	15th Jan 26
Accepted:	30th Jan 26

Editor: Esther Schnapp

Transaction Report:

Dear Dr. Xu,

Thank you very much for your patience while your manuscript was peer-reviewed at EMBO reports. We have now received the full set of referee reports that are pasted below.

As you will see, the referees acknowledge that the findings are interesting. However, they also have several suggestions for how the study can be improved and strengthened and I think all suggestions are good and should be addressed. Please let me know in case you have any questions or comments and we can discuss the exact revision requirements further, also in a video chat, if you like.

I would thus like to invite you to revise your manuscript with the understanding that the referee concerns must be fully addressed and their suggestions taken on board. Please address all referee concerns in a complete point-by-point response. Acceptance of the manuscript will depend on a positive outcome of a second round of review. It is EMBO reports policy to allow a single round of major revision only and acceptance or rejection of the manuscript will therefore depend on the completeness of your responses included in the next, final version of the manuscript.

We realize that it is difficult to revise to a specific deadline. In the interest of protecting the conceptual advance provided by the work, we recommend a revision within 3 months (10th Jan 2026). Please discuss the revision progress ahead of this time with the editor if you require more time to complete the revisions.

- 1) A data availability section providing access to data deposited in public databases is missing. If you have not deposited any data, please add a sentence to the data availability section that explains that.
- 2) Your manuscript contains statistics and error bars based on $n=2$. Please use scatter blots in these cases. No statistics should be calculated if $n=2$.

3) We replaced Supplementary Information with Expanded View (EV) Figures and Tables that are collapsible/expandable online. A maximum of 5 EV Figures can be typeset. EV Figures should be cited as 'Figure EV1, Figure EV2' etc... in the text and their respective legends should be included in the main text after the legends of regular figures.

5) a complete author checklist, which you can download from our author guidelines <https://www.embopress.org/page/journal/14693178/authorguide>. Please insert information in the checklist that is also reflected in the manuscript. The completed author checklist will also be part of the RPF.

6) Please note that all corresponding authors are required to supply an ORCID ID for their name upon submission of a revised manuscript (<https://orcid.org/>). Please find instructions on how to link your ORCID ID to your account in our manuscript tracking system in our Author guidelines <https://www.embopress.org/page/journal/14693178/authorguide#authorshipguidelines>

10) Regarding data quantification (see Figure Legends:

<https://www.embopress.org/page/journal/14693178/authorguide#figureformat>)

12) All Materials and Methods need to be described in the main text using our 'Structured Methods' format, which is required for all research articles. According to this format, the Methods section includes a separate Reagents and Tools Table file (listing key reagents, experimental models, software and relevant equipment and including their sources and relevant identifiers) and a Methods and Protocols section describing the methods using a step-by-step protocol format. The aim is to facilitate adoption of the methodologies across labs. More information on how to adhere to this format as well as a downloadable template (.docx) for the Reagents and Tools Table can be found in our author guidelines:

An example of a Method paper with Structured Methods can be found here: <https://www.embopress.org/doi/full/10.1038/s44320-024-00037-6#sec-4>

You are able to opt out of this by letting the editorial office know (emboreports@embo.org). If you do opt out, the Review

Process File link will point to the following statement: "No Review Process File is available with this article, as the authors have chosen not to make the review process public in this case."

I look forward to seeing a revised form of your manuscript when it is ready.

Referee #1:

The Xu's laboratory further investigated telomere uncapping using the temperature-sensitive *cdc13-1* allele. They found that genomic rearrangements occur at telomeres and subtelomeres after a transient depletion of Cdc13 protein. Nicely, long-read sequencing allowed to precisely map these rearrangements of the clones that escape from telomere uncapping. In the current scientific context, this is an incremental work that sheds light on some aspects of telomere instability, most notably the importance of t-circle-mediated-amplification in type II-like survivors.

As mentioned in the introduction, Cdc13 is a part of CST (Cdc13-Stn1-Ten1) complex that fulfills crucial functions at telomeres: protection, elongation, c-strand synthesis and replication. To my point of view, the authors did not carefully consider the possibility that telomeres and subtelomeres rearrangements might be driven by replication stress after Cdc13 depletion rather than uncapping only. Here are some comments that I believe could strengthen this study.

Comments:

-It might be worth mentioning that *cdc13-1* is a temperature-sensitive mutant and also an hypomorph mutant (DOI: 10.1534/genetics.111.137869)

- For greater clarity, it would be beneficial to show a figure with X and Y telomeres, type I and type II survivors including Xho1 sites to facilitate the analysis of TRF southern blots.

-Analysis of figure 2B is tricky since several clones do not exhibit changes compared to WT. Since the sequencing was not performed on each clone (Fig 3B), hybridization with a Y' probe could facilitate the examination of the TRF Southern blot.

-page 89 lane 2: "bands of different intensities", it rather looks like a difference of sample loading or it should be clearly shown on the gel of figure 2A.

-The choice of clones for sequencing or telomere length analysis is not fully justified (Figure 3B,4B...). Figure 4B, why not C3, C12 and C13.

- A main concern is the possibility that rearrangements are the consequence of perturbed replication of chromosome ends when Cdc13 is dysfunctional at 32°C. To show that YAS and T-II-L are issued from telomere uncapping only, cells should be blocked in G2 before temperature shift, then analysis of survivors should be done. This is an important point that may change the interpretation of the results. This may explain why the effect of Rad52 is only partial (to be discussed). Along the same line, I was wondering what would be the consequence of the absence of Exo1. Did the authors test it since they suggest that extensive resection should occur?

-page 17 lane 19-21: This is confusing. Why survivors to Cdc13 depletion would partially depend on Rad52 if rearrangements strongly depend on Rad52. This calls into question the use of the term "survivor".

-The absence of Pol32 seems to favor T-II-L (Figure 5A). Is this statistically relevant? This might reflect the need of telomerase when replication is challenged.

-page 14 lane 19: Telomere lengthening in type II survivors in telomerase-negative cells is thought to be attributed to rolling circle amplification. However, it has not been clearly demonstrated and results from figure S2D is a strong argument to support this hypothesis. This point could be emphasized more strongly. How do the authors explain the presence of t-circles? Are there t-circles naturally present in *cdc13-1* mutant? Is that this point can be addressed?

Minor points:

-Figure S2A: indicate the name of clones for each lane and add the WT control to allow comparison

Referee #2:

In the manuscript entitled „transient telomere uncapping triggers telomeric and subtelomeric rearrangements" the authors have used the well characterized temperature sensitive yeast mutant, *cdc13-1*, to induce telomere dysfunction in a controlled manner. The authors start by demonstrating that uncapping telomeres only leads to cell death, the inability to re-grow at permissive temperature, after 6 hours of growth at the non-permissive temperature of 32{degree sign}C. They set out to determine what type of genomic re-arrangements may occur following prolonged telomere uncapping. Using PFGE they could see that indeed many chromosomes were altered in terms of their migration pattern, suggestive of larger chromosomal aberrations. To test whether rearrangements were occurring in the telomeric/subtelomeric region, Southern blots were performed. The authors report 2 observations from the capping survivors, some of the clones had acquired a heterogeneous telomere length, similar to type II survivors, which they referred to as type-II-like (T-II-L) and the remaining survivors either had altered Y' or X patterns, which they referred to as Y'-associated survivors (YAS). Using longread nanopore sequencing, the authors were able to determine that indeed, Y' elements were both lost and gained following telomere uncapping. Importantly, using this approach, it could be determined that all rearrangements were confined to telomere-proximal regions.

Through careful analysis of the sequences, they were able to determine that the addition of telomere length was independent of telomerase, but more likely a recombination mediated event, through the copying of circles through BIR. Consistently, the deletion of either *rad52*, or *pol32* abolished the T-II-L and the YAS phenotypes, as assayed by Southern blotting. Interestingly, the T-II-L became resistant to further decapping events, a phenotype that was dependent on the presence of *RAD52*.

All in all, this manuscript is very clear and adds considerable knowledge to the understanding of what events occur when telomeres become uncapped. I only have very minor comments, but otherwise I am very supportive of this being published in EMBO reports and I think it is of high interest to the telomere field especially.

1. I feel that figure 1 and 2 could be potentially combined into a single figure.
2. Although it has been shown thousands of times, it would be very beneficial for the non-telomere biologists to show a diagram of the yeast telomeres (X and Y) with the restriction sites and showing where the probe hybridizes. It makes the Southern blots much more accessible.
3. I find it interesting that the T-II-L clones become resistant to a second telomere uncapping. What happens if these cells lose telomerase? Do they bypass senescence?

Referee #3:

Yeast telomeres are capped by several proteins including *Cdc13* - a component of the CST (*Cdc13/Stn1/Ten1*) complex - which binds to the G-overhang. In this manuscript, Liebaut Dudragne and co-authors use the extensively described *cdc13-1* temperature-sensitive allele to investigate the consequences of transient telomere uncapping and the structure of the telomeres in the rare cells that resume growth when returned to permissive temperature. Long-read sequencing provides here a number of observations that complement previous publications on capping mutants. These results will be of interest to yeast researchers and, more broadly, the field of telomere instability, including research on ALT cancers, which depends on recombination for telomere maintenance.

Interestingly, although telomere recombination is commonly observed following transient telomere uncapping, it does not appear to be a prerequisite, as survivors are formed even in the absence of *Rad52*. Telomere recombination includes a wide range of rearrangements of the subtelomeric regions, as well as telomere elongation. Unlike survivors from telomerase inactivation, only a few telomeres in each clone exhibit rearrangements. This suggests that most telomeres remain functional and that recombination is stochastic. Signatures of rolling circle amplification from t-circles, identifiable as perfect tandem repeats, are detected by long-read sequencing. Although this mechanism has long been invoked to explain telomere elongation, to my knowledge this is the most compelling argument yet for the use of this mechanism *in vivo*.

The manuscript is well written, the experimental conditions are clearly explained and the message is clear. The manuscript contains a large amount of data of excellent quality. However, it feel that more information could be provided in the supplemental (see below).

Here are few comments/questions:

- Fig. 1 shows a drastic decrease in recovery after 24 hours at restrictive temperature compared to 12 hours. Does survival after 12 hours also depend partially on recombination, or is recombination a consequence of extensive incubation at restrictive

temperature?

- Lydall's lab identified mutations affecting the growth of *cdc13-1* cells at various temperatures (Holstein et al., G3, 2017). Did the authors check for any mutations that could have been selected in the survivors?
- Without quantification of the Y' sequences using a subtelomeric-specific probe, the Southern blots are not very informative as compared to the sequencing and the commentary is confusing. For example, in Fig2A, clones C4, C5, C9 and C10 have a similar profile to the control, despite the text stating that 'almost all other survivors showed visible alterations in Y'-specific band intensity'. The same is true for the fig5 A-D. The Figure S1 does not include a control.
- The figure 3B could be more detailed. Perhaps an additional version could be included in the supplemental, showing subcategories of the 'modified Y' structure (loss, gain, amplification and new). Similarly, Fig. 3D clearly illustrates the changes at some telomeres. Would it be possible to extend this representation to cover more, or even all, telomeres in the supplementary data?
- Figs 4D and S2D show the rolling circle amplification signature in type II-like survivors. This section could benefit from more detail. Are sequences of different lengths independent of each other? How often is the same repeat found on several telomeres? Can a single telomere contain several types of repeat? Does their sequence allow the original telomere to be identified? Are tandem repeats observed in the absence of RAD51 (clone C17 Fig5E)?
- Fig. S2D shows two examples in which the repeats are located at the beginning of the telomeres. Is this a typical trait that would suggest rolling circle is an initial process in forming type II-like telomeres? Further examples could be provided to illustrate the position of the tandem repeats.
- As expected, the fraction of remodeled telomeres in the double *rad51 rad59* mutant is similar to that in *rad52* cells (Fig5F). Nevertheless, the recovery frequency is higher for *rad51 rad52* cells (Fig S3A). Could you comment?

Minor points:

- A schematic of the *S. cerevisiae* telomeres would help non-yeast scientists to understand the telomere rearrangement described in the paper.
- In Figure 3D, TEL01R in the control contains two Y' elements, whereas SGD describes TEL01R as a X-only telomere in the reference strain. Is this a mistake or are there variations from strain to strain? If so, the structure of the 32 telomeres of the strain used in this study should be included in the supplementary data, to make the figures easier to interpret.
- Figure S4 is entitled 'Type II-like survivors arrest in G2/M in response to telomere uncapping'. However, the graph shows fractions of G1 cells. While I agree that the decrease in the fraction of G1 cells most probably correlates with an increase in G2/M arrested cells, this is not strictly correct. G2/M arrest is better characterized by DAPI staining as a % of mononucleate large budded cells. This should be corrected in both the text and figure legend.
- Page 25, lane 17: 'Type II survivors are strongly increased in the absence of RAD51'. Please add a reference.
- Page 26, lane 5: Though both refs (Aguilera et al., 2022 and Larrivee and Wellinger, 2006) clearly showed the presence of t-circles in type II survivors, none of them strictly proved their use for telomere elongation.
- The origin of t-circles should be discussed, since they have only been detected in survivors of telomerase inactivation up to now.
- Page 26, lanes 20-24: Another difference is the presence of active telomerase upon release of the *cdc13-1* cells at permissive temperature.

Point-by-point response to reviewers' comments and editorial requests

(Our responses are shown in blue)

Editorial comments:

1) A data availability section providing access to data deposited in public databases is missing. If you have not deposited any data, please add a sentence to the data availability section that explains that.

The data availability section provides a link to the European Nucleotide Archive (<https://www.ebi.ac.uk/ena/browser/home>) where all our sequencing data (reads and assemblies) have been deposited with the project accession PRJEB93986.

2) Your manuscript contains statistics and error bars based on $n=2$. Please use scatter blots in these cases. No statistics should be calculated if $n=2$.

The genome assembly results concerning the *rad52Δ* mutant were indeed based on $n = 2$ independent sequenced survivor strains. We have now sequenced two more, thus reaching $n = 4$.

Referee #1:

The Xu's laboratory further investigated telomere uncapping using the temperature-sensitive *cdc13-1* allele. They found that genomic rearrangements occur at telomeres and subtelomeres after a transient depletion of Cdc13 protein. Nicely, long-read sequencing allowed to precisely map these rearrangements of the clones that escape from telomere uncapping. In the current scientific context, this is an incremental work that sheds light on some aspects of telomere instability, most notably the importance of t-circle-mediated-amplification in type II-like survivors.

As mentioned in the introduction, Cdc13 is a part of CST (Cdc13-Stn1-Ten1) complex that fulfills crucial functions at telomeres: protection, elongation, c-strand synthesis and replication. To my point of view, the authors did not carefully consider the possibility that telomeres and subtelomeres rearrangements might be driven by replication stress after Cdc13 depletion rather than uncapping only. Here are some comments that I believe could strengthen this study.

We wish to thank the reviewer for the constructive feedback that helped improve our manuscript.

Comments:

-It might be worth mentioning that *cdc13-1* is a temperature-sensitive mutant and also an hypomorph mutant (DOI: 10.1534/genetics.111.137869)

We have now indicated this information when we introduce the allele (p.4 lines 16-17).

- For greater clarity, it would be beneficial to show a figure with X and Y telomeres, type I and type II survivors including Xho1 sites to facilitate the analysis of TRF southern blots.

We now include a schematic representation of the restriction fragments visualized in the Southern blot, at the bottom of Fig. 2B. Using the same representation, the end structures of YAS and type-II-like survivors are shown in Fig. 3D.

-Analysis of figure 2B is tricky since several clones do not exhibit changes compared to WT. Since the sequencing was not performed on each clone (Fig 3B), hybridization with a Y' probe could facilitate the examination of the TRF Southern blot.

We have reprobbed the Southern blot in Fig. 2B with a Y' probe (new Fig. EV1B) and evidence modest changes in band intensity. We changed the text on p. 7 lines 17-20 accordingly and more specifically refer to "visible loss of X fragments, and/or alterations in Y'-fragments intensity when marked with a Y'-specific probe". We also replaced "almost all other" by "the majority of other". Importantly, clones 3, 5, 26 and 29, which do not show obvious TRF Southern blot profile changes, contain numerous rearrangements as evidenced by nanopore sequencing (Fig. 3B). Overall, taking into account the TRF Southern blots probed by telomeric or Y' probes, and the sequencing data, we conclude that the majority of the surviving clones show rearranged subtelomeres or telomeres.

-page 89 lane 2: "bands of different intensities", it rather looks a difference of sample loading or it should be clearly be shown on the gel of figure 2A.

Indeed, the band intensities of different clones cannot be compared directly. We actually meant "relative intensities" of the chromosome bands within a clone (we have now modified the text on p. 8 line 5). We have now added a set of green arrows in Fig. 2A to explicitly indicate the bands that display these altered relative intensities, or blurry bands, indicative of intraclonal heterogeneity, which is then demonstrated in Fig. 2C for clone 3 and in Fig. EV1C for clones 2 and 5.

-The choice of clones for sequencing or telomere length analysis is not fully justified (Figure 3B,4B...). Figure 4B, why not C3, C12 and C13.

YAS survivors were randomly selected for sequencing, but we tried to include both altered (c2, c17) and unaltered/slightly altered (c3, c5, c26, c29) Southern blot profiles.

For type-II-like clones, we selected the clones from our second Southern blot (Fig. EV1A) because long telomeric bands were better resolved, allowing for possible correlation of these bands with Nanopore-sequencing-derived telomeric lengths, although we did not perform this analysis in the end.

This justification has been added to the text on p. 9 lines 6-8.

- A main concern is the possibility that rearrangements are the consequence of perturbed replication of chromosome ends when Cdc13 is dysfunctional at 32°C. To show that YAS and T-II-L are issued from telomere uncapping only, cells should be blocked in G2

before temperature shift, then analysis of survivors should be done. This is an important point that may change the interpretation of the results. This may explain why the effect of Rad52 is only partial (to be discussed).

Along the same line, I was wondering what would be the consequence of the absence of Exo1. Did the authors test it since they suggest that extensive resection should occur?

It is well established that *cdc13-1* cells incubated at restrictive temperature accumulate single-stranded DNA specifically at telomere/subtelomere regions and in a mostly Exo1-dependent manner (Garvik et al. 1995 PMID: 7565765; Maringele and Lydall 2002 PMID: 12154123; Zubko et al. 2004 PMID: 15454530). We have now performed a survival assay in a *cdc13-1 exo1Δ* strain incubated transiently at 32°C and found that, in line with these previous works, the survival frequency is increased by 50-100 folds compared to *cdc13-1 EXO1*. We have added this result in Fig. EV3B.

Therefore, it is most likely that the rearrangements are caused telomere deprotection and subsequent Exo1-dependent accumulation of single-stranded DNA. It is still possible that the disruption of CST's function in assisting replication by C-strand fill-in might also contribute to a certain extent to single-strand DNA accumulation or other source of genomic stress. In this work we remain agnostic to the exact molecular mechanisms at the origin of the rearrangements, apart from showing that survival is greatly increased in the *exo1Δ* mutant (Fig. EV3B).

We still tried to perform the experiment suggested by the reviewer. Cells were arrested in G2 by depletion of Cdc20, using an integrated *pGAL-CDC20* construct (as we did previously in Coutelier et al. 2023, PMID: 36342193). Cells were first grown at 23°C in galactose-containing media, then arrested for 5h by glucose addition and transferred to 32°C for 24h. After wash to eliminate the glucose and plating on galactose-medium (to induce *CDC20* expression again), cells were allowed to grow at 23°C. The colonies that formed were then analyzed by TRF Southern blot (see Fig_review 1).

Fig_review 1. TRF Southern blot of 15 survivor clones after the following experimental protocol: *cdc13-1 pGal-CDC20* cells were arrested in G2 in liquid culture by glucose addition, incubated for 24h at 32°C, and plated on a galactose-containing plate to allow recovery at 23°C.

We found that some colonies still displayed YAS profiles with altered X-bands (clones 8, 9, 12 and 15) and we did not detect type-II-like profiles. Although the frequency of YAS appeared to be decreased compared to Fig. 2B, we note that the protocol had to be modified to perform this experiment: cells could not be plated on a glucose-containing plate at 32°C because we would not have been able to reactivate *CDC20* on the same plate. Thus, we had to add glucose and incubate the cells at 32°C in liquid for 24h. However, we noticed that the culture grew substantially in liquid, which we did not expect. We suspect that that some genetic or phenotypic suppressor cells resistant to the *CDC20* arrest or to the *cdc13-1* deprotection managed to grow and outcompete the rest of the population. In addition, the 24h of G2 arrest likely impacted cell survival, further biasing the results. Thus, this experiment was not informative and it can be experimentally challenging to properly address the contribution of the replication functions of CST to the rearrangements using this type cell-cycle arrest approach.

We note that when we performed the transient uncapping experiment with an adaptation-deficient mutant *cdc5-ad*, which stays fully checkpoint-arrested during the 24h (see Coutelier et al. 2023, PMID: 36342193), we still evidence many rearrangements (Fig. EV1D), indicating that replication during telomere uncapping (through adaptation to DNA damage) is not required for the rearrangements. We performed a TRF Southern blot analysis of *cdc13-1 cdc5-ad* clones surviving the transient uncapping and detected 5 out of 16 type-II-like profiles (See Fig_review 2). YAS profiles could not be assessed due to poor hybridization of the X bands unfortunately. However, we nanopore-sequenced several *cdc13-1 cdc5-ad* clones and did find Y' element rearrangements (not shown), consistent with the altered PFGE profiles (Fig. EV1D).

Fig_review 2. TRF Southern blot of 16 *cdc5-ad* survivor clones using the standard assay.

Overall, we believe that a rigorous examination of a potential implication of a telomere replication defect would require much more work and would go beyond the scope of this paper, which aims at characterizing the downstream rearrangements. However, since we cannot rule out that a replication defect at telomeres caused by *cdc13-1* might contribute to some extent to the phenotypes we observe, we now mention this possibility in the discussion (p. 18 lines 15-21).

This may explain why the effect of Rad52 is only partial (to be discussed).

The effect of Rad52 on rearrangements is not partial but close to total, since only one Y' alteration remains in 4 *rad52Δ* survivor clones (we sequenced 2 more clones for this revised version), as compared to 71 in 9 *RAD52* survivors (Fig. 5E and Fig. 3B).

-page 17 lane 19-21: This is confusing. Why survivors to Cdc13 depletion would partially depend on Rad52 if rearrangements strongly depend on Rad52. This calls into question the use of the term "survivor".

The fact the *rad52Δ* cells show a survival frequency that is only two folds less than wild-type while no longer displaying rearrangements suggests that rearrangements are a consequence of telomere uncapping but are not strictly required for survival. At best, they can contribute to a better survival, consistently with our observation that the type-II-like survivors are resistant to a second uncapping.

We have now modified the mentioned sentence to clarify this point (now p. 14 line 24 to p. 15 line 2).

-The absence of Pol32 seems to favor T-II-L (Figure 5A). Is this statistically relevant? This might reflect the need of telomerase when replication is challenged.

In the absence of *POL32*, we did not detect any type-II-like pattern in Southern blot. The *pol32Δ* mutant did show increased telomere length (as previously described in Gatbonton et

al. 2006 PMID: 16552446), both in the control strain at 23°C and in some survivors (Fig. 5A and Fig. EV4B), but it is nowhere close the massive telomere elongation of up to several kilobases found in type-II-like survivors.

-page 14 lane 19: Telomere lengthening in type II survivors in telomerase-negative cells is thought to be attributed to rolling circle amplification. However, it has not been clearly demonstrated and results from figure S2D is a strong argument to support this hypothesis. This point could be emphasized more strongly. How the authors explain the presence of t-circles? Is there t-circles naturally present in *cdc13-1* mutant? is that this point can be addressed?

We thank the reviewer for suggesting to emphasize this aspect of our work. We now provide more analyses and details, including exhaustive maps of t-circle-derived repeats for all sequenced T-II-L survivors in the new Fig. 4E, Fig. EV2D-E, Fig. EV4E. The text in the results section has been modified accordingly on p. 13 (and p. 15 lines 19-22 for *rad51Δ*). We also discuss the origins of the t-circle (p. 20 lines 5-12).

Our analysis of tandem repeat pattern in Fig. 4D did not reveal any significant signature of t-circles in the control *cdc13-1* strain. Some tandem repeats of 2 copies of sequences of <45 bp are detected but are likely due to recombination.

We also refer to this response to a similar question by reviewer #3:

Since the telomere motif in *S. cerevisiae* is imperfect, we can assess whether the sequences found in tandem repeats are different, i.e. most likely independent, or not. We found that most circle sequences were independent, with the exception of Circle_139 in clone 21, whose sequence is embedded in the sequence of Circle_332 (and of Circle_58 in the *rad51Δ* T-II-L survivor). We could trace the sequence of Circle_139 back to the telomere of chromosome V-L in the genome of the control strain. Then, Circle_332 likely stemmed from a secondary excision event after Circle_139 was first copied onto another telomere. We could also find the origin of one circle in the *rad51Δ* T-II-L clone 17, but not for the other circles, likely because the terminal telomere sequences experience shortening and elongation by telomerase, which would add unique sequences and erase the previous ones. Finally, a single telomere can contain repeats associated with different circles.

Minor points:

-Figure S2A: indicate the name of clones for each lane and add the WT control to allow comparison

We have now indicated the name of each clone on this TRF Southern blot. Because it was performed for screening additional clones, the blot did not contain the usual control strain unfortunately. To better visualize the alteration of the band profile, we added a separate control next to it (Fig. EV2A).

Referee #2:

In the manuscript entitled „transient telomere uncapping triggers telomeric and subtelomeric rearrangements" the authors have used the well characterized temperature sensitive yeast mutant, *cdc13-1*, to induce telomere dysfunction in a controlled manner. The authors start by demonstrating that uncapping telomeres only leads to cell death, the inability to re-grow at permissive temperature, after 6 hours of growth at the non-permissive temperature of 32°C. They set out to determine what type of genomic re-arrangements may occur following prolonged telomere uncapping. Using PFGE they could see that indeed many chromosomes were altered in terms of their migration pattern, suggestive of larger chromosomal aberrations. To test whether rearrangements were occurring in the telomeric/subtelomeric region, Southern blots were performed. The authors report 2 observations from the capping survivors, some of the clones had acquired a heterogeneous telomere length, similar to type II survivors, which they referred to as type-II-like (T-II-L) and the remaining survivors either had altered Y' or X patterns, which they referred to as Y'-associated survivors (YAS). Using longread nanopore sequencing, the authors were able to determine that indeed, Y' elements were both lost and gained following telomere uncapping. Importantly, using this approach, it could be determined that all rearrangements were confined to telomere-proximal regions.

Through careful analysis of the sequences, they were able to determine that the addition of telomere length was independent of telomerase, but more likely a recombination mediated event, through the copying of circles through BIR. Consistently, the deletion of either *rad52*, or *pol32* abolished the T-II-L and the YAS phenotypes, as assayed by Southern blotting. Interestingly, the T-II-L became resistant to further decapping events, a phenotype that was dependent on the presence of *RAD52*.

All in all, this manuscript is very clear and adds considerable knowledge to the understanding of what events occur when telomeres become uncapped. I only have very minor comments, but otherwise I am very supportive of this being published in EMBO reports and I think it is of high interest to the telomere field especially.

We thank the reviewer for their very positive feedback.

1. I feel that figure 1 and 2 could be potentially combined into a single figure.

The two figures could indeed have been fused, but now that we have added a schematic representation of the restriction fragments of the Southern blot, Fig. 2 is larger and we think we should keep them separated.

2. Although it has been shown thousands of times, it would be very beneficial for the non-telomere biologists to show a diagram of the yeast telomeres (X and Y) with the restriction sites and showing where the probe hybridizes. It makes the Southern blots much more accessible.

We now include a schematic representation of the restriction fragments visualized by the Southern blot, at the bottom of Fig. 2B.

3. I find it interesting that the T-II-L clones become resistant to a second telomere uncapping. What happens if these cells lose telomerase? Do they bypass senescence?

We passaged T-II-L survivors for 11 passages and observed progressive telomere shortening in the presence of telomerase (Fig. EV5B), although the telomeres remained very long. Makovets and colleagues (Makovets et al. 2008 PMID: 18202371) found that for type II survivors in which telomerase was reintroduced, telomere length was not fully re-equilibrated even after ~300 generations. Most likely, telomerase acted very little on these very long telomeres since telomerase has a preference toward short telomeres (Teixeira et al. 2004 PMID: 15109493). We therefore assumed that in our case, even without telomerase, it would also take many more passages to decrease the average telomere length to the initial length and then test whether senescence crisis would occur or not. It would also be plausible that, if a telomerase gene (say *TLC1* or *EST2*) is deleted in any of these survivors, the very long telomeres could poise these cells toward telomerase-independent type II survivor phenotype, thus immediately bypassing senescence. In any case, while very interesting, all these experiments would have taken at least several dozens of passages and we deemed them outside the scope of this work on transient telomere deprotection.

Referee #3:

Yeast telomeres are capped by several proteins including Cdc13 - a component of the CST (Cdc13/Stn1/Ten1) complex - which binds to the G-overhang. In this manuscript, Liebaut Dudragne and co-authors use the extensively described *cdc13-1* temperature-sensitive allele to investigate the consequences of transient telomere uncapping and the structure of the telomeres in the rare cells that resume growth when returned to permissive temperature. Long-read sequencing provides here a number of observations that complement previous publications on capping mutants. These results will be of interest to yeast researchers and, more broadly, the field of telomere instability, including research on ALT cancers, which depends on recombination for telomere maintenance.

Interestingly, although telomere recombination is commonly observed following transient telomere uncapping, it does not appear to be a prerequisite, as survivors are formed even in the absence of Rad52. Telomere recombination includes a wide range of rearrangements of the subtelomeric regions, as well as telomere elongation. Unlike survivors from telomerase inactivation, only a few telomeres in each clone exhibit rearrangements. This suggests that most telomeres remain functional and that recombination is stochastic. Signatures of rolling circle amplification from t-circles, identifiable as perfect tandem repeats, are detected by long-read sequencing. Although this mechanism has long been invoked to explain telomere elongation, to my knowledge this is the most compelling argument yet for the use of this mechanism *in vivo*.

The manuscript is well written, the experimental conditions are clearly explained and the message is clear. The manuscript contains a large amount of data of excellent quality.

We thank the reviewer for the encouraging comments.

However, it feel that more information could be provided in the supplemental (see below).

Here are few comments/questions:

- Fig. 1 shows a drastic decrease in recovery after 24 hours at restrictive temperature compared to 12 hours. Does survival after 12 hours also depend partially on recombination, or is recombination a consequence of extensive incubation at restrictive temperature?

We have now quantified the survival to a 12-hour transient telomere deprotection in wild-type, *rad52Δ* and *pol32Δ* cells (now in Fig. EV3B). The wild-type cells showed a ~20 fold decrease in survival compared to the "0 hour" control, consistent with the spot assay shown in Fig. 1B. Interestingly, the deletion of *RAD52* did not significantly affect this survival frequency whereas the deletion of *POL32* did.

This result is consistent with the idea that Rad52-dependent telomere and subtelomere rearrangements are a consequence of transient telomere uncapping and with 12 hours of telomere deprotection, survival is likely driven simply by checkpoint arrest and cell cycle restart.

The result in *pol32Δ* cells is more intriguing and is possibly linked to a yet unexplored genetic interaction between *POL32* and *cdc13-1* or *CDC13*. We also noticed that *pol32Δ* cells displayed a slight growth defect at 23°C, which might have some unforeseen effect in combination with telomere uncapping. The text has been modified on p. 14 lines 8-10 to include this result.

- Lydall's lab identified mutations affecting the growth of *cdc13-1* cells at various temperatures (Holstein et al., G3, 2017). Did the authors check for any mutations that could have been selected in the survivors?

We thank the reviewer for this suggestion. We have now analyzed the genomes of the survivor clones and identified all variants using variant caller Clair3. We then compared the variants affecting the same gene in at least two strains to the lists from three genetic screens from the Lydall lab: Holstein et al. 2017, Addinall et al. 2008 (PMID: 18845848) and Addinall et al. 2011 (PMID: 21490951). Only one affected gene was found in two YAS survivors, *MTC7*, which was listed also in Addinall et al. 2008. However, in that work, the *MTC7* mutant increased the growth defect of *cdc13-1* at restrictive temperature and is therefore not selected in our survivors.

We added a paragraph in the results section (p. 9 lines 12-21) and in the Methods section.

- Without quantification of the Y' sequences using a subtelomeric-specific probe, the Southern blots are not very informative as compared to the sequencing and the commentary is confusing. For example, in Fig2A, clones C4, C5, C9 and C10 have a similar profile to the control, despite the text stating that 'almost all other survivors showed visible

alterations in Y'-specific band intensity'. The same is true for the fig5 A-D. The Figure S1 does not include a control.

Compared to long-read sequencing, the TRF Southern blots allowed us to assess a larger number of surviving clones. We have now added a Southern blot probing the Y' elements in Fig. EV1B and indeed observed some mild changes in Y' band intensity, consistent with the quantification based on sequencing (Fig. 3C). Of note, in clone 9, an X band around 5kb is missing (Fig. 2B). We have now rephrased the corresponding text and more specifically refer to "visible loss of X fragments, and/or alterations in Y'-fragments intensity when marked with a Y'-specific probe" (p. 7 lines 17-19). We also replaced "almost all other" by "the majority of other".

In Fig. 5A-D, there are much less visibly altered bands, especially in *rad52Δ*, *pol32Δ* and *rad51Δrad59Δ*, which is also consistent with the sequencing results.

The Southern blot in Fig. EV1A did not contain the usual control strain unfortunately, as it was performed only to screen more surviving clones. To better visualize the alteration of the band profile, we added a separate control next to it (Fig. EV1A).

- The figure 3B could be more detailed. Perhaps an additional version could be included in the supplemental, showing subcategories of the 'modified Y' structure (loss, gain, amplification and new). Similarly, Fig. 3D clearly illustrates the changes at some telomeres. Would it be possible to extend this representation to cover more, or even all, telomeres in the supplementary data?

We thank the reviewer for the suggestion. We have now added a version in Fig. EV3C showing, for all strains, more subcategories: gain (amplifications are included in "gain"), loss and new.

The Appendix Data S2 (previously Supp. Data 2) contains the representation of all chromosome extremities for all strains and all sequenced clones.

- Figs 4D and S2D show the rolling circle amplification signature in type II-like survivors. This section could benefit from more detail. Are sequences of different lengths independent of each other? How often is the same repeat found on several telomeres? Can a single telomere contain several types of repeat? Does their sequence allow the original telomere to be identified? Are tandem repeats observed in the absence of RAD51 (clone C17 Fig5E)?

Since the telomere motif in *S. cerevisiae* is imperfect, we can assess whether the sequences found in tandem repeats are different, i.e. most likely independent, or not. We found that most circle sequences were independent, with the exception of Circle_139 in clone 21, whose sequence is embedded in the sequence of Circle_332, and of Circle_58 in the *rad51Δ* T-II-L survivor. We could trace the sequence of Circle_139 back to the telomere of chromosome V-L in the genome of the control strain. Then, Circle_332 likely stemmed from a secondary excision event after Circle_139 was first copied onto another telomere. We could also find the origin of one circle in the *rad51Δ* T-II-L clone 17, but not for the other circles, likely because the terminal telomere sequences experience shortening and

elongation by telomerase, which would add unique sequences and erase the previous ones. Finally, a single telomere can contain repeats associated with different circles.

All these more detailed analyses are now included in the text p. 13 (and p. 15 lines 19-22 for *rad51Δ*) and in the new Fig. 4E, Fig. EV2D-E, Fig. EV4E) where we provide maps of the circle-derived repeats on all chromosome ends.

- Fig. S2D shows two examples in which the repeats are located at the beginning of the telomeres. Is this a typical trait that would suggest rolling circle is an initial process in forming type II-like telomeres? Further examples could be provided to illustrate the position of the tandem repeats.

We now provide an exhaustive map of the repeated circle-related sequences for the 3 T-II-L clones (Fig. 4E and Fig. EV2D-E), as well as the one in the *rad51Δ* strain (Fig. EV4E). These sequences can be found at any position of the telomeres.

- As expected, the fraction of remodeled telomeres in the double *rad51 rad59* mutant is similar to that in *rad52* cells (Fig5F). Nevertheless, the recovery frequency is higher for *rad51 rad52* cells (Fig S3A). Could you comment?

The difference in survival frequency between *rad52Δ* and *rad51Δ rad59Δ* is not statistically different. Therefore, *rad52Δ* and *rad51Δ rad59Δ* are indistinguishable, both at the survival frequency level and regarding rearrangements.

Minor points:

- A schematic of the *S. cerevisiae* telomeres would help non-yeast scientists to understand the telomere rearrangement described in the paper.

We now include a schematic representation of the telomere/subtelomere regions in *S. cerevisiae* as well as the restriction fragments visualized by the Southern blot, at the bottom of Fig. 2B.

- In Figure 3D, TEL01R in the control contains two Y' elements, whereas SGD describes TEL01R as a X-only telomere in the reference strain. Is this a mistake or are there variations from strain to strain? If so, the structure of the 32 telomeres of the strain used in this study should be included in the supplementary data, to make the figures easier to interpret.

Indeed, Y' elements vary from strain to strain (see O'Donnell et al. 2023 PMID: 37524789 or our preprint Dudragne et al. 2025 bioRxiv 2025.11.13.688250 for a comprehensive description of Y' element diversity across *S. cerevisiae* strains).

The subtelomeric structures of all 32 chromosome extremities of the control strain, as well as all sequenced mutants and survivor clones, are already shown in Appendix Data S2 (previously Supp. Data 2).

- Figure S4 is entitled 'Type II-like survivors arrest in G2/M in response to telomere

uncapping'. However, the graph shows fractions of G1 cells. While I agree that the decrease in the fraction of G1 cells most probably correlates with an increase in G2/M arrested cells, this is not strictly correct. G2/M arrest is better characterized by DAPI staining as a % of mononucleate large budded cells. This should be corrected in both the text and figure legend.

We have now corrected the figure legend and the corresponding text (p. 17 lines 1-2).

- Page 25, lane 17: 'Type II survivors are strongly increased in the absence of RAD51'. Please add a reference.

We have now added the reference Teng et al. 2000 PMID: 11090632.

- Page 26, lane 5: Though both refs (Aguilera et al., 2022 and Larrivee and Wellinger, 2006) clearly showed the presence of t-circles in type II survivors, none of them strictly proved their use for telomere elongation.

We now write that in both studies, t-circles were "detected" in telomerase-negative type II survivors (p. 20 line 9).

- The origin of t-circles should be discussed, since they have only been detected in survivors of telomerase inactivation up to now.

We now mention in the discussion that t-circles might emerge through excision of an endogenous telomere sequence and that they could originate from the same mechanisms that were proposed to operate in telomerase-negative type II survivors, although these are still not fully elucidated either (p. 20 lines 6-12).

- Page 26, lanes 20-24: Another difference is the presence of active telomerase upon release of the cdc13-1 cells at permissive temperature.

We have now added a sentence about the presence of telomerase during and after telomere uncapping (p. 21 lines 7-9).

Dear Dr. Xu,

Thank you for the submission of your revised manuscript. We have now received the enclosed reports from the referees that were asked to assess it, and I am happy to say that both support its publication now. Only a few editorial requests will need to be addressed before we can proceed with the official acceptance of your manuscript:

- Please correct the conflict of interest subheading to "Disclosure and Competing Interests Statement"
- The author credits need to be removed from the ms file. All credits need to be entered during online ms submission.
- The REFERENCE format is not correct: et al needs to be used after 10 author names; DOIs should only be used for preprints and datasets that have not been published yet. Please correct to the EMBO reports format.
- In the author checklist, the section on statistics has not been completed. Please send us a new, completed checklist.
- The two Data files uploaded as Appendix Data 1 and 2 need to be corrected to Dataset EV1 and EV2 in all places (source file names, titles in the system, callouts in the ms) and uploaded as 2 individual Dataset files (so they are not part of the Appendix). The legends need to be removed from the Appendix file and provided as a text file in each Dataset zip folder.
- The 2 tables in the Appendix file can be uploaded as Table EV1 and Table EV2. The legends for the tables need to be in the table files. In this way, the Appendix file can be deleted/removed.
- In the Data Availability Section, we need the specific URLs for the large deposited datasets.
- The Methods section needs to be labeled as 'Methods'
- The section with EV figure legends needs to be labeled as 'Expanded View Figure Legends'
- Please note that the Abstract needs to be written in present tense.

* Figure Legends - Comments *

- Please note that the exact p values are not provided in the legends of figures 5F, EV3 A, B. Please provide exact values as reasonable.
- Please note that the box plots need to be defined in terms of minima, maxima, centre, bounds of box and whiskers, and percentile in the legends of figures 4C, EV2 A-C
- Please note that information related to n is missing in the legends of figures 4C, EV2 A-C
- Please note that the error bars are not defined in the legends of figures EV5 A.

EMBO press papers are accompanied online by A) a short (1-2 sentences) summary of the findings and their significance, B) 2-3 bullet points highlighting key results and C) a synopsis image that is exactly 550 pixels wide and 200-600 pixels high (the height is variable). The synopsis image should provide a sketch of the major findings, like a graphical abstract. Please note that text needs to be readable at the final size. Please send us this information along with the final manuscript.

Referee #1:

This is a significant and well-executed revision that clearly improves upon the initial version. I fully acknowledge the authors' efforts and the thorough work invested in addressing all the concerns raised by the referees.

Referee #3:

The authors did a great work addressing or discussing my concerns and suggestions. The manuscript is now is very solid, and the conclusions are very well supported by the results. Therefore, I am fully supportive of acceptance.

The authors have addressed all minor editorial requests.

Dr. Zhou Xu
Sorbonne Université - CNRS
Laboratory of Computational, Quantitative and Synthetic Biology
Sorbonne Université Institut de Biologie Paris Seine Laboratoire de Biologie Computationnelle et Quantitative - UMR7238
Telomere and Genome Stability Group
4 place Jussieu, Bâtiment C, 3ème étage
Paris, - 75252
France

Dear Dr. Xu,

I am very pleased to accept your manuscript for publication in the next available issue of EMBO reports. Thank you for your contribution to our journal.

You may qualify for financial assistance for your publication charges - either via a Springer Nature fully open access agreement or an EMBO initiative. Check your eligibility: <https://link.springer.com/journal/44319/how-to-publish-with-us>

Yours sincerely,

>>> Please note that it is EMBO Reports policy for the transcript of the editorial process (containing referee reports and your response letter) to be published as an online supplement to each paper. If you do NOT want this, you will need to inform the Editorial Office via email immediately. More information is available here: <https://link.springer.com/partners/embo-press/editorial-policies#Peer%20review>